# Correlative single molecule lattice light sheet imaging reveals the dynamic relationship between nucleosomes and the local chromatin environment

Timothy A. Daugird[1,7], Yu Shi[2,7], Katie L. Holland [3], Hosein Rostamian[4,5], Zhe Liu [3], Luke D. Lavis [3], Joseph Rodriguez[6], Brian D. Strahl [4,5] & Wesley R. Legant [1,2] ✉

In the nucleus, biological processes are driven by proteins that diffuse through and bind to a meshwork of nucleic acid polymers. To better understand this interplay, we present an imaging platform to simultaneously visualize single protein dynamics together with the local chromatin environment in live cells. Together with super-resolution imaging, new fluorescent probes, and biophysical modeling, we demonstrate that nucleosomes display differential diffusion and packing arrangements as chromatin density increases whereas the viscoelastic properties and accessibility of the interchromatin space remain constant. Perturbing nuclear functions impacts nucleosome diffusive properties in a manner that is dependent both on local chromatin density and on relative location within the nucleus. Our results support a model wherein transcription locally stabilizes nucleosomes while simultaneously allowing for the free exchange of nuclear proteins. Additionally, they reveal that nuclear heterogeneity arises from both active and passive processes and highlight the need to account for different organizational principles when modeling different chromatin environments.

The nucleus is a heterogeneous environment that is functionally and physically partitioned at multiple length scales. This partitioning occurs through spatial variations in the concentrations of nucleic acids and proteins within the nuclear space and allows the nucleus to perform diverse functions including DNA replication[1], transcription[2,3], RNA splicing[4], and ribosome biogenesis[5]. At the most fundamental scale, 147 DNA base pairs wrap around an octamer of histone proteins, collectively called a nucleosome[6]. Nucleosomes coalesce into heterogeneous groups of ~4-15 clusters or 'clutches' at a scale of 10's of nm[7,8].

These clutches further aggregate into irregular chromatin nanodomains containing 1000's of base pairs that partition at the length scale of 100-200 nm and are separated by an interchromatin space that is enriched in RNA and transcriptional machinery[9–11]. At the scale of the entire genome, individual chromosomes occupy distinct regions of the nucleus referred to as chromosome territories[12,13].

Computational and physical models provide insight into the thermodynamic principles by which chromatin organization is established (reviewed in[14]); however, a challenge remains to determine how

[1]Department of Pharmacology, University of North Carolina at Chapel Hill, Chapel Hill, NC, USA. [2]Joint Department of Biomedical Engineering, University of North Carolina at Chapel Hill, North Carolina State University, Chapel Hill, NC, USA. [3]Janelia Research Campus, Howard Hughes Medical Institute, Ashburn, VA 20147, USA. [4]Department of Biochemistry and Biophysics, University of North Carolina at Chapel Hill, Chapel Hill, NC, USA. [5]Curriculum in Genetics and Molecular Biology, University of North Carolina at Chapel Hill, Chapel Hill, NC, USA. [6]National Institute of Environmental Health Sciences, Durham, NC 27709, USA. [7]These authors contributed equally: Timothy A. Daugird, Yu Shi. ✉e-mail: legantw@email.unc.edu

variations in chromatin density within the nucleus relate to the functional specification of different nuclear processes. Single molecule imaging has shown that nucleosomes within nanodomain cores move coherently and are less mobile than those existing elsewhere in the nucleus[10,15]. Nucleosomes at the nuclear periphery, within dense chromocenters, or in differentiated cells (vs. pluripotent cells), all of which are enriched in heterochromatin, move less over time and are more radially confined than those in euchromatin[10,16]. Such findings suggest that dense heterochromatin regions may be more crowded and less physically accessible to nuclear proteins than sparse euchromatin regions[17]. Intriguingly, active processes appear to alter nucleosome motion in different ways. Transcription and DNA looping appear to stabilize nucleosomes[10,18,19] whereas DNA damage repair destabilizes them[18]. Together, these findings reveal a dynamic interplay between different nuclear processes, chromatin density, and the thermodynamically driven motion of nucleosomes. However, recent reports have also reported that nucleosome motion is independent of chromatin density, on average, when investigating replication inhibited cells with the same chromatin content, but double the nuclear volume compared to control cells[20] and other studies have demonstrated what even dense heterochromatin regions are equally accessible to an inert probe like green fluorescent protein[21].

Because of these discrepancies, we sought to better determine the relationship between chromatin density, nucleosome motion, the physical properties of the interchromatin space, and specific nuclear functions. Toward this goal, we developed an imaging platform that combines live-cell 3D single molecule tracking (SMT) together with high-resolution volumetric imaging via lattice light sheet microscopy[22,23]. We used this platform to simultaneously visualize nucleosome motion in the context of local chromatin density variations in the nucleus. We combined these measurements together with fixed-cell 3D super-resolution imaging of nucleosome packing, diffusion measurements of inert fluorescent probes, targeted perturbations of specific nuclear functions, and biophysical modeling of chromatin organization.

Our findings demonstrate that nucleosome motion, spatial organization, and sensitivity to pharmacological and genetic perturbations all vary as a function of the local chromatin density in the nucleus. Intriguingly, the viscoelastic properties of the interchromatin space appear to be constant in both sparse and dense chromatin environments, suggesting that the observed differences in nucleosome behavior are more likely attributed to active processes, such as transcription, that locally stabilize nucleosomes in sparse euchromatic regions. Overall, our results provide a window into a heterogeneous and dynamic nuclear environment and provide an avenue to incorporate this heterogeneity into future models of chromatin function, spatial organization, and dynamics.

## Results

### Lattice light sheet microscopy enables simultaneous 3D tracking of individual nucleosomes together with high-resolution measurement of local chromatin density

We utilized a fibroblast-like Cos7 cell line that stably over-expresses histone H2B fused to a self-labeling HaloTag (Cos7-Halo-H2B)[24,25]. Previous reports indicate that exogenously expressed HaloTag-H2B is uniformly integrated into the mammalian genome[18] and co-labeling of our Cos-Halo-H2B cells with HaloTag-Ligand Janelia Fluor 549 and Hoechst indicates that HaloTag-H2B integrates into sparse and dense chromatin with similar propensity (Supplementary Fig. 1A). Quantification via western blot indicated that HaloTag-H2B was expressed at roughly 4.4% compared to the endogenous protein (Supplementary Fig. 1B). To enable the simultaneous observation of individual nucleosomes and the surrounding chromatin microenvironment, we developed a two-color labeling and imaging protocol with lattice light sheet microscopy (Fig. 1A, See methods). We performed live-cell 3D

single molecule imaging with photoactivatable JaneliaFluor-646 to track nucleosomes with a lateral precision of $24 \pm 9$ nm and axial precision of $137 \pm 59$ nm (mean ± std, Fig. 1B, C, Supplementary Fig. 1C) and simultaneous diffraction-limited volumetric imaging with Janelia Fluorphre 525 to resolve chromatin density with $334 \times 837$ nm resolution (full width half maximum) laterally and axially (Fig. 1C). In this scenario, each ($110 \times 110 \times 250$ nm) voxel of the diffraction limited image could consist of ~90,000 base pairs of DNA and 625 nucleosomes. This assumes that nucleosomes are optimally packed within the volume (Fig. 1D) and it is roughly comparable to the 500 nm estimate interaction radius of HiC[26].

To isolate our SMT analysis specifically to DNA-bound H2B, we classified localizations as belonging to the same trajectory if they moved <400 nm over the course of sequential 20 ms frames. This approach allowed us to focus exclusively on the behavior of DNA-bound H2B and effectively filter out trajectories that may be due to free diffusing H2B or non-specifically bound dye molecules. To quantify chromatin density, we used an expectation maximization algorithm to segment the chromatin images into distinct classes based on intensity[27] (Fig. 1E), hereafter referred to as chromatin density classes (CDCs). Partitioning chromatin into seven CDCs provided a robust fit to the intensity histograms while not overfitting the data according to the Bayesian information criterion and Akaike information criterion (Supplementary Fig. 1D). We found that simultaneous imaging of the underlying chromatin micro-environment was essential. Even at ~5.5 sec/volume sampling, the underlying chromatin organization was quite dynamic with individual voxels transitioning through the entire range of CDC values within a time course (Supplementary Fig. 1E, Supplementary Movie 1). Finally, to investigate the relationship between chromatin density and nucleosome motion, we classified each nucleosome trajectory according to the underlying CDC and fit the cell ensemble mean square displacement (MSD) to a model of anomalous diffusion (Fig. 1F, Supplementary Movie 2). We note that the CDCs represent a statistical partitioning of a smooth intensity distribution rather than describing physically distinct chromatin regions. Nevertheless, the segmented chromatin images allowed us to quantify the relative amounts and spatial arrangement of dense vs. sparse chromatin in the nucleus and to partition nucleosome trajectories into similar chromatin densities across different cells.

### Single nucleosomes display differential motion based on local chromatin micro-environment

We found that, on average, nucleosomes in denser CDCs display a slower apparent diffusion coefficient (Fig. 2A, B). This trend was consistent across eight biological replicates (Supplementary Fig. 2A) and showed significant differences in nucleosome motion across CDCs (two-sided Spearman coefficient = −0.344, p-value < 1E-5, pairwise one-sided $t$-tests in Supplementary Fig. 2B). Interestingly, we found that the negative correlation between chromatin density and nucleosome motion was strongest for the nucleosome diffusion coefficient and the radius of gyration (two-sided Spearman coefficient = −0.48, $p$ < 1E-5) (Fig. 2B, Supplementary Fig. 2C). Nucleosome displacement anisotropy displayed a smaller, though still significant correlation (two-sided Spearman coefficient = −0.10, $p$ = 0.006) (Supplementary Fig. 2D), while the anomalous diffusion exponent displayed no significant differences across CDCs (Fig. 2C) (one-way ANOVA F-statistic = 0.31, $p$ = 0.93). This is further supported by analysis in which CDC's were computationally scrambled in a manner that shifted their locations but preserved their overall distribution and gross morphological features (Supplementary Fig. 3A–C). Under these conditions, we found no significant correlation between nucleosome dynamics and the spatially scrambled CDCs (one-way ANOVA F-statistic = 0.21, $p$ = 0.97) (Supplementary Fig. 3D). The observed differences in diffusion coefficients are notably larger than the combined precision of detection and tracking, as evidenced by the lack of correlation

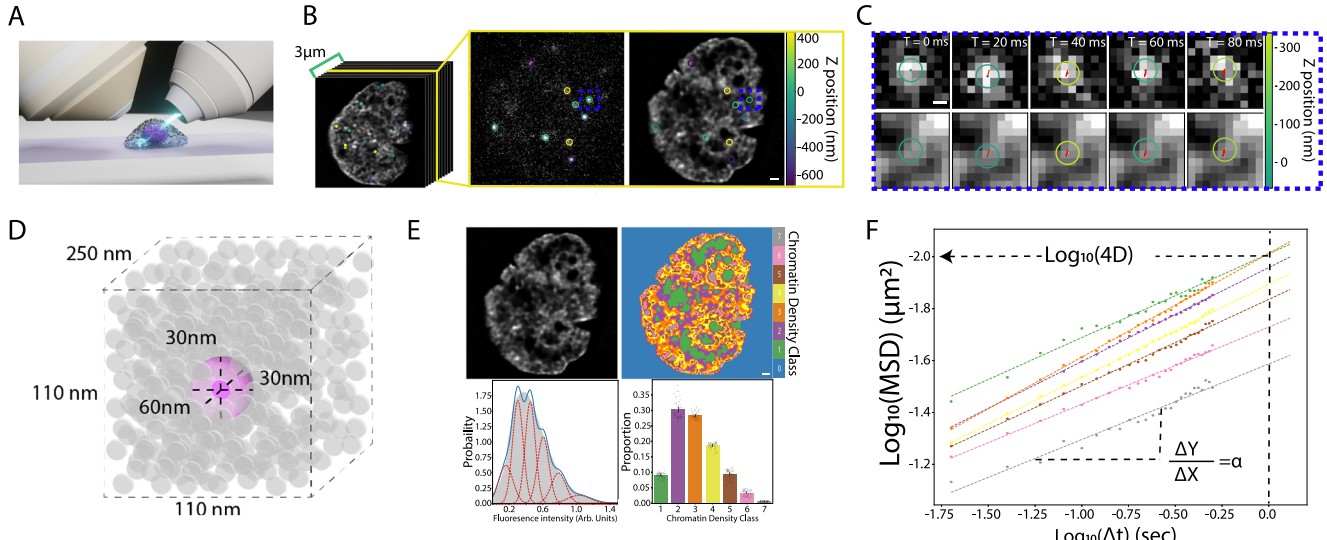

**Fig. 1 | Correlative single nucleosome imaging. A** Schematic of lattice light sheet microscopy. **B** A sample slice of single nucleosomes (middle) and their associated chromatin environment (right) from a 3D volume of cell nucleus (left). Circles indicate tracked nucleosomes. Colors represent the z position relative to the focal plane. Scale bar = 1000 nm **C** The trajectory of the nucleosome in the blue box in (**B**). Top: Single nucleosome trajectory tracked across five consecutive 20 ms frames; Bottom: Overlay of nucleosome trajectory on the chromatin channel. **D** Schematic comparing lattice light sheet microscopy voxel size (dashed cube), nucleosome localization precision (magenta sphere), and the potential nucleosome density (gray disks). **E** Chromatin density classification. Top left: deconvolved chromatin image; Top right: corresponding chromatin density classification; Bottom left: histogram of chromatin intensity in arbitrary units for top deconvolved

image. Blue curve indicates combined Gaussian Mixture Model fitting and red curves indicate individual components; Bottom right: histogram of the proportion of voxels assigned to different chromatin density classes for cell displayed in top image. Scale bar = 1000 nm. Bar height indicates mean and error bars represent standard deviation. Dots represent proportion from a single time point. $N = 1$ biological replicate taken across 50 consecutive time points. **F** Representative mean square displacement of nucleosomes in log-log scale for a single cell. The dashed lines show the linear fitting to a power law relationship (MSD = $4D\Delta t^{\alpha}$) where $\alpha$, $D$ and $\Delta t$ are the anomalous alpha exponent, the apparent diffusion coefficient respectively and the time lag. Colors indicate different chromatin density classes with the same convention as in (**E**).

(one-way ANOVA F-statistic = 0.24, p = 0.96) between the nucleosome diffusion coefficient and chromatin density class after paraformaldehyde fixation (Supplementary Fig. 3D). In total, these findings demonstrate that nucleosomes in denser chromatin environments exhibit slower diffusion compared to those in sparser chromatin environments and indicates a link between chromatin density and nucleosome dynamics, with the diffusion coefficient and radius of gyration serving as key parameters.

We hypothesized that the observed differences in nucleosome motion across CDCs may arise from various factors, including differences in the local viscoelastic properties of the material surrounding chromatin, variations in nucleosome spatial packing, and/or differences in active processes that may be localized to a given CDC. These factors, either individually or in combination, could contribute to the distinct patterns of nucleosome dynamics observed in different CDCs. Subsequently, our investigation aimed to identify and characterize the biophysical properties and biological processes responsible for these observed differences in nucleosome motion.

### The interchromatin space displays similar viscoelastic properties regardless of chromatin density

We next sought to investigate the relationship between chromatin density and the viscoelastic properties of the interchromatin space. To this end, we extended our two-color labeling and imaging protocol to track the motion of a non-interacting HaloTag fused to a nuclear localization signal (HaloTag-NLS) as it diffused within the nucleus (Fig. 2D, Supplementary Movie 3). For each CDC, we calculated the cell-ensemble MSD for free-diffusing HaloTag-NLS and extracted the apparent diffusion coefficient and anomalous alpha exponent and through fitting a power law model. To reduce linking errors for fast-moving particles, we deliberately maintained a lower density of HaloTag-NLS localizations compared to the density of nucleosome

localizations. This resulted in fewer trajectories at higher chromatin density levels, which could potentially cause a systematic shift in the population median. Consequently, we determined the minimum number of trajectories necessary for a reliable estimation of diffusion parameters and excluded any line fits from cells with fewer than 250 trajectories in a given condition (Supplementary Fig. 3E).

In contrast to our nucleosome measurements and largely consistent with previous work performing correlative imaging of freely diffusing dextrans[21], we found no apparent dependence in either the apparent diffusion coefficient or the anomalous alpha exponent on chromatin density classes (one-way ANOVA F-statistic = 1.07, $p = 0.38$ for diffusion coefficient, and F-statistic = 0.84, $p = 0.54$ for anomalous alpha exponent) (Fig. 2E, F). This indicates that, for a non-interacting free diffusing particle of ~2.6 nm size[28], different chromatin density classes display similar viscoelastic properties. We also found that the anomalous alpha exponent is about 0.8, which is close to what one would expect for diffusion in purely viscous liquid (anomalous alpha exponent of one).

### Nucleosomes in dense chromatin environments are more optimally space filling and randomly packed than nucleosomes in sparse chromatin environments

To measure the spatial arrangement of nucleosomes in different CDC's we utilized 3D single molecule localization microscopy. Specifically, we labeled cells with a stochastically blinking HaloTag-Ligand Janelia-Fluor630B probe[29,30], chemically fixed them with paraformaldehyde, and imaged them on a custom built highly inclined swept tile (HIST) microscope similar to that described in[31] (Fig. 3A). Because photo-toxicity is not important for fixed-cell imaging, we chose HIST because of its larger field of view parallel to the coverslip compared to lattice light sheet microscopy. This arrangement allowed us to localize individual nucleosomes to $18 \pm 8$ nm laterally and $80 \pm 28$ nm axially

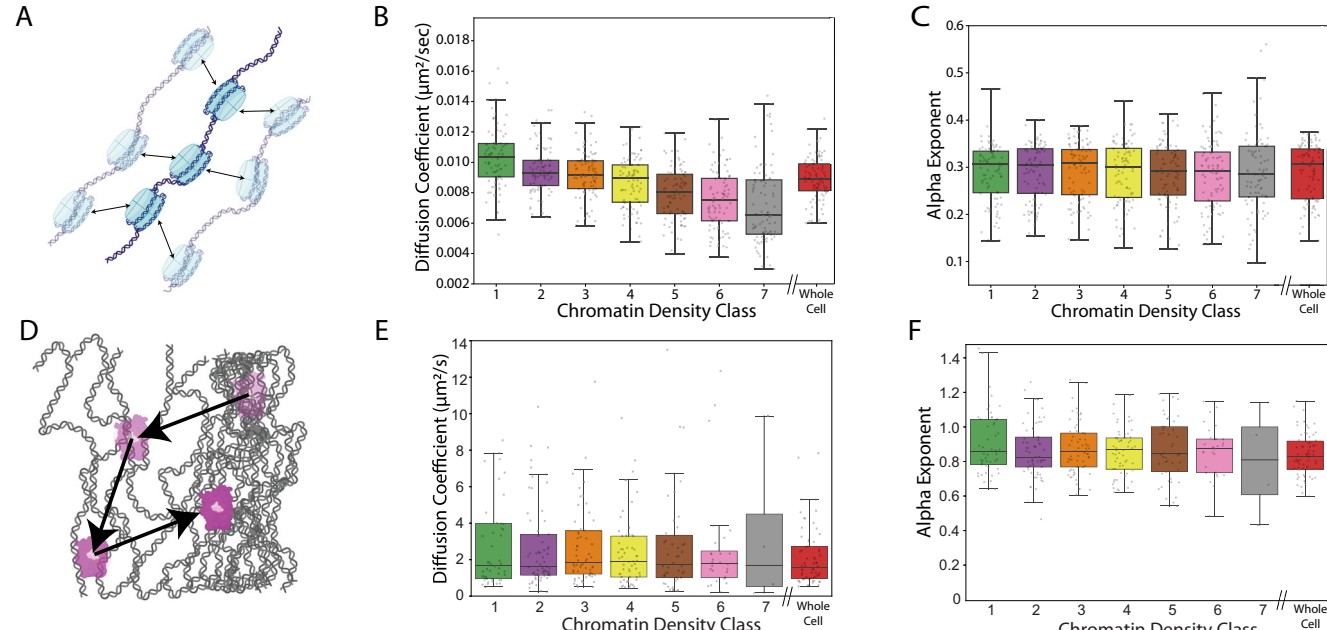

**Fig. 2 | Nucleosome dynamics and their associated chromatin density. A** A schematic of nucleosome diffusion. **B** Box plot of the nucleosome diffusion coefficient across different chromatin classes. The box indicates the inter- quartile range. The horizontal line in the middle indicates median, and the bars indicate 1.5 x the upper and lower limits of inter-quartile range. Dots indicate fitted values from a single cell. **C** Box plot of the nucleosome anomalous alpha exponent. The plot follows the same convention as (**B**). **D** A schematic of HaloTag-NLS diffusion. **E** Box plot of the diffusion coefficient for HaloTag-NLS across different chromatin density classes. The plot follows the same convention as (**B**). **F** Box plot of the extracted anomalous alpha exponent for HaloTag-NLS across different chromatin density classes. The plot follows the same convention as (**B**). Data from (**B**) and (**C**) are from $n = 88$ cells across 8 independent biological replicates. Data from (**E**) and (**F**) are from $n = 37$ cells across 4 independent biological replicates. Box plots in (**B**), (**C**), (**E**), and (**F**) include only points from line fits to data that include at least 250 trajectories.

(mean ± std, Supplementary Fig. 4A). To establish a comparable comparative framework to our live cell imaging, we convolved the 3D localization density histogram (Fig. 3B) with an experimentally measured lattice light sheet point spread function, resulting in a simulated diffraction-limited image (Fig. 3C). This simulated image then underwent the same processing pipeline and chromatin density classification as our live cell images (Fig. 3D, E). Subsequently, each individual nucleosome localization was classified based on the underlying CDC (Fig. 3F, Supplementary Movie 4). Comparing the power spectrum of our simulated diffraction-limited images with that of our live cell images revealed quantitatively similar morphological features (Supplementary Fig. 4B). Furthermore, the distributions of the CDCs in our simulated images closely resembled those observed in our live cell imaging (Supplementary Fig. 4C). These findings indicate that our processing and classification pipeline successfully identifies and characterizes comparable chromatin density regions in both the super-resolution data and live cell imaging.

To quantify nucleosome organization within each CDC, we computed the normalized 3D pair correlation function G(r), (Fig. 3G) which quantifies the deviation from spatial randomness at a given length scale[32]. Unlike other methods for determining spatial clustering such as DBSCAN[33,34] or tesselation-based approaches[35,36], G(r) does not invoke implicit assumptions about discrete cluster sizes or cluster number. Furthermore, because G(r) is normalized by an equal number of randomly distributed points occupying the same space, it corrects for edge effects from masking and decouples the nucleosome spatial arrangement from variations in nucleosome density[32]. Consistent with previous reports[37], G(r) measurements of nucleosomes across the entire nucleus indicated linear scaling regimes, indicative of clustering along a continuum of spatial scales. The scaling can be characterized by two continuous power law like regimes: the first extends from the

limit of our precision (~30 nm) to 186 nm ± 48 (mean ± std), and the second continues from there to 1088 ± 270 nm (Fig. 3H, Supplementary Fig. 4D). This power law like behavior is consistent with a fractal-like organization of nucleosomes, with the power law exponent corresponding to a fractal dimension[38]. Fitting the linear regimes of our log-log plotted data yields estimated fractal dimensions of 2.72 ± 0.06 for the lower regime and 2.91 ± 0.02 for the upper regime (Supplementary Fig. 4D). These estimates are consistent with previous reports[37] and orthogonal methods for determining the fractal dimension of chromatin[21,39,40].

Interestingly, we found that, in addition to localization density (Fig. 3I), the fractal dimension of nucleosome packing was also positively correlated with chromatin density (two-sided Spearman correlation coefficient = 0.78, Fig. 3J, K). The lowest density CDC 1 had a fractal dimension of 2.14 ± 0.25 whereas the highest density CDC 7 had a fractal dimension of 2.85 ± 0.09. The largest jump was observed between CDCs 1 and 2 (one-sided t-statistic = −23.89, $p < 1E\text{-}5$) with a progressive increase in fractal dimension between CDCs 2 through 7 (two-sided Spearman coefficient = 0.67, $p < 1E\text{-}5$). When we repeated this analysis in scrambled CDCs, we found no clear differences in localization density (one-way ANOVA F-statistic = 0.94, $p = 0.46$ or G(r) curves (Supplementary Fig. 4E, F)). These deviations from random organization were also clearly seen in representative reconstructions of the nucleosome localizations in different CDCs (Supplementary Fig. 5).

Overall, these results demonstrate that the spatial arrangement of nucleosome packing varies together with the local chromatin density in the nucleus. Nucleosomes in low density CDCs display a lower fractal dimension, are more clustered, and less randomly distributed than nucleosomes in high density CDCs which are more randomly distributed and more optimally space filling.

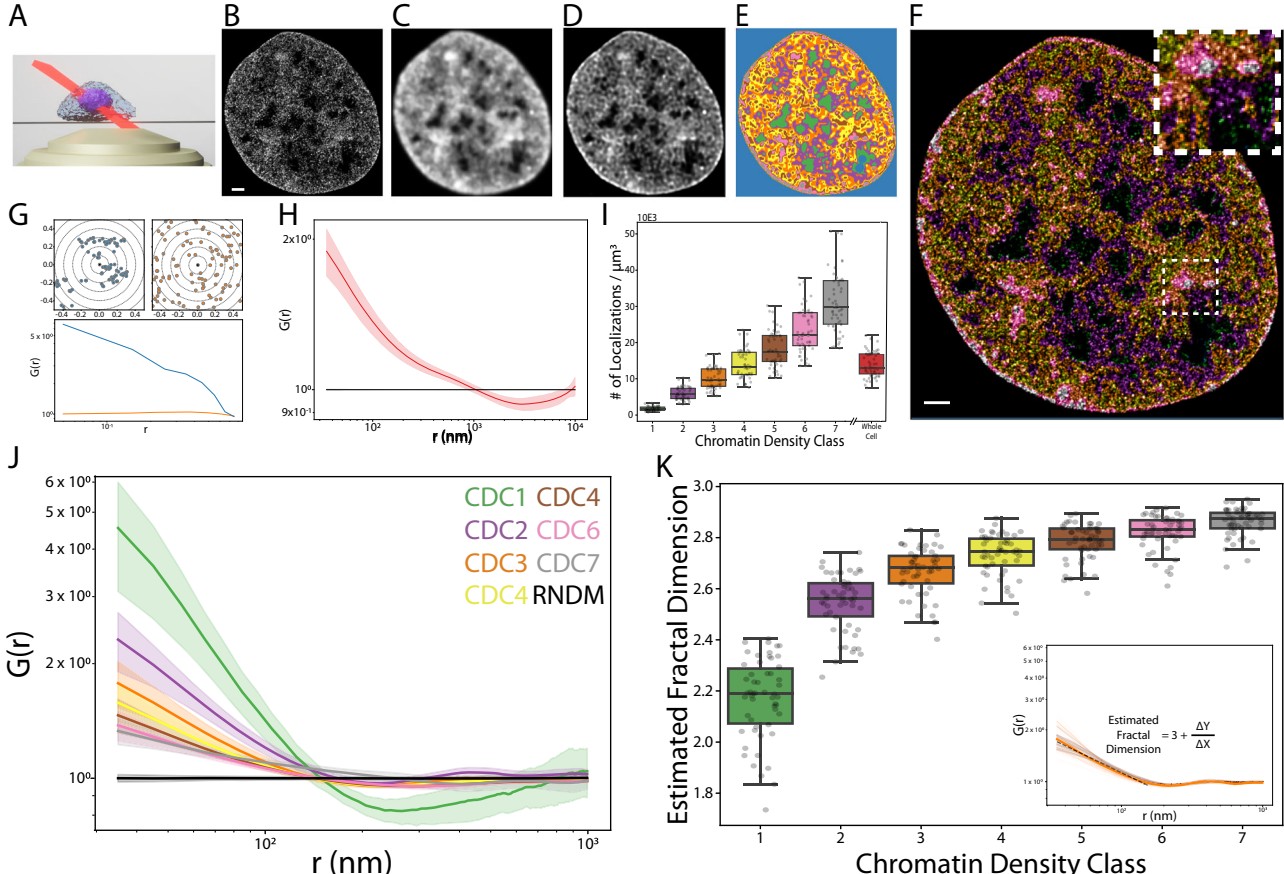

**Fig. 3 | Measurements of nucleosome organization in different chromatin density classes. A** Schematic of highly inclined swept tile imaging for single molecule localization of nucleosome organization. **B–E** Representative images for the processing steps used to assign chromatin density classes to super resolution nucleosome images. The super resolution reconstruction (**B**) is convolved with an experimental lattice light sheet microscopy point spread function to generate a simulated diffraction-limited image (**C**). This image is then deconvolved (**D**) and segmented into chromatin density classes (**E**). Scale bar in (**B**) = 1000 nm and applies to (**B–E**). **F** Individual nucleosomes are assigned chromatin density classification according to their localization in (**E**). The top right dashed box in (**F**) is the zoom-in of the ROI – 2.3 × 2.3 μm. **G** Demonstration of the pair correlation function G(r) for a simulated clustered distribution (blue) and a simulated uniform distribution (orange). **H** Pair correlation function of the nucleosome organization for

the entire cell (red) and a completely random distribution (black). Solid line indicates mean value of all cells and shaded region indicates standard deviation. **I** Box plot of localization density in different chromatin classes. The plot follows that same convention as Fig. 2B. **J** Pair correlation function of the nucleosome organization for different chromatin density classes (CDCs). The black curve is the G(r) curve for a random distribution (RNDM). Solid lines indicate mean value of all cells and shaded regions indicated standard deviation. **K** Box plot of the estimated fractal dimension for different chromatin classes. The plot follows the same convention as (**I**). Inset shows example power law fitting (black dashed line) to the G(r) curve for single chromatin density class in a single cell (dark orange). Opaque lines represent G(r) curves for all other cells. Data from (**I–K**) are from $n = 54$ cells across 3 independent biological replicates.

## The nuclear periphery represents a distinct biophysical environment that affects nucleosome dynamics and spatial organization

To more completely understand the distinct behaviors observed in the nucleus interior and edge, it is crucial to consider that, apart from increased chromatin density at the nuclear periphery, the laminal region is distinctly characterized by its attachment to the inner nuclear membrane[41–43] and repression of transcription[44,45]. To more completely understand how these factors affect nucleosome dynamics, we partitioned the nucleus according to a distance from the edge of our nuclear mask (Fig. 4A).

We observed a significant positive correlation between nucleosome diffusion coefficient and the distance from the nuclear edge in the outermost 500 nm of the nucleus (one-sided $t$-test for the regression coefficient, $p < 1E$-5) (Fig. 4B). Intriguingly, the anomalous alpha exponent of nucleosome motion also shows a similar correlation within these regions, (one-sided $t$-test for the regression coefficient, $p < 1E$-5) (Fig. 4C), whereas it is unaffected by local chromatin density (Fig. 2C). Contrary to an assumption of uniformly dense chromatin at

the nuclear periphery, our findings reveal that it is a heterogeneously structured chromatin region, akin to the rest of the nucleus. This is evidenced by the widespread distribution of CDCs throughout the nucleus, with a modest increase in high-density CDCs near the periphery (Fig. 4D). These observations suggest that a general description of the nuclear periphery as an area of densely compacted chromatin does not fully capture the unique characteristics of the nuclear edge that contribute to the more anomalous motion of nucleosomes.

In order to further probe unique aspects of nuclear organization at the edge, we next segmented our CDC images into 'nuclear periphery' and 'nuclear interior' according to a cutoff of 500 nm from the edge of the nuclear mask (Fig. 4E). We found that nucleosomes in the nuclear interior displayed comparable class dependent differences in diffusion coefficient as our whole nucleus data (two-sided Spearman coefficient = −0.51, $p = < 1E$-5) (Fig. 4F, gray). Interestingly, although nucleosomes, on the average, show slower diffusion at the nuclear periphery compared to the nuclear interior, the dependence between nucleosome motion and chromatin density is almost entirely lost at the nuclear periphery (Fig. 4F, white). While nucleosomes in periphery

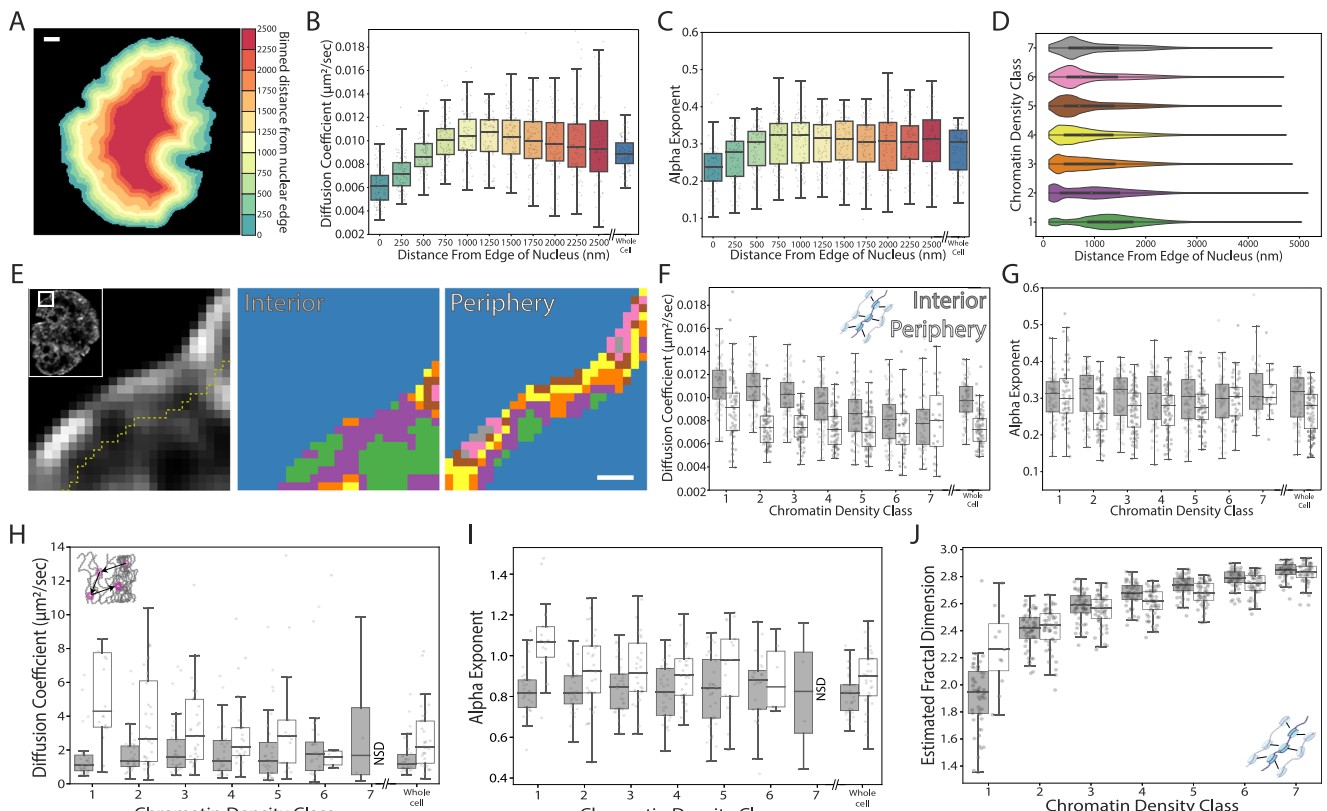

**Fig. 4 | Differences in nucleosome dynamics and organization at nuclear periphery and nuclear interior. A** Example distance to nuclear edge image. Scale bar = 1000 nm (**B**) Box plot of the diffusion coefficient as function of distance from nuclear edge. The plot follows the same convention as Fig. 2B. **C** Box plot of the anomalous alpha exponent as a function of distance from nuclear edge. The plot follows the same convention as (**B**). **D** Violin plot of voxel distance from the nuclear edge within each chromatin density class. Boxes indicates the inter-quartile range. White dot indicates median, and bars indicate 1.5 x the limits of inter-quartile range. **E** (Left) Example deconvolved image. Dashed line indicates 500 nm from the nuclear edge. (Middle, Right) Example CDC images at the nuclear interior and periphery. Scale bar = 500 nm (**F**) Box plot of the diffusion coefficient across different chromatin classes at the nuclear interior (Gray) and nuclear periphery (White). The plot follows the same convention as (**B**). **G** Box plot of the anomalous alpha exponent, the plot follows the same color assignments and convention as (**F**).

**H** Box plot of the diffusion coefficient for HaloTag-NLS across different chromatin density classes at the nuclear interior and nuclear periphery. The plot follows the same color assignments and convention as (**F**). **I** Box plot of the anomalous alpha exponent for HaloTag-NLS across different chromatin density classes at the nuclear interior and nuclear periphery. The plot follows the same color assignments and convention as (**F**). **J** Box plot of the estimated fractal dimension across different chromatin density classes at the nuclear interior and nuclear periphery. The plot follows the same color assignments and convention as (**F**). Data from (**B**–**D**), (**F**), and (**G**) are from $n = 88$ cells across 8 independent biological replicates. Data from (**H**) and (**I**) are from $n = 37$ cells across 4 independent biological replicates. Data from (**J**) are from $n = 54$ cells across 3 independent biological replicates. Box plots in (**F**–**I**) include only points from line fits to data that include at least 250 trajectories, and "NSD" indicates not sufficient data. Box plot in (**J**) includes only points from line fits with $r^2 > 0.95$.

CDC1 display a higher diffusion coefficient compared to the other periphery CDCs, we found no differences in nucleosome diffusion coefficient in periphery CDC2-7 (ANOVA F-Statistic, 0.57, $p = 0.72$) (Fig. 4F). Moreover, while we observed no significant variation in the anomalous alpha exponent across the CDCs for the nuclear interior, we observe significant variations in the anomalous alpha exponent across all CDCs at the nuclear periphery, with the most pronounced differences in the intermediate CDCs 2–4 (Fig. 4G). Importantly, we observe no significant differences in the diffusion coefficient, or anomalous alpha exponent in scrambled CDCs at the nuclear edge or nuclear interior (Supplementary Fig. 6A, B). Collectively, these findings suggest that the nuclear periphery represents a unique environment that places distinct constraints on nucleosome motion, and that nucleosomes in similarly compacted chromatin regions at the periphery vs. the nuclear interior display unique diffusive behaviors.

We next sought to probe the possible variation in viscoelastic properties between nuclear periphery and interior via the motion of HaloTag-NLS. Interestingly, while HaloTag-NLS motion was uncorrelated with CDC in the nuclear interior, we found that HaloTag-NLS displayed slower, more anomalous motion through the denser CDCs at the nuclear edge (two-sided Spearman coefficient = −0.24, $p = 1E$-3 for

diffusion coefficient, two-sided Spearman coefficient = −0.21, $p = 5E$-3 for anomalous alpha exponent) (Fig. 4H, I). Moreover, the diffusion coefficient at nuclear periphery was significantly higher than interior in CDC1 and CDC2 (pairwise $T$-test, $p = 1E$-4 and $p = 1E$-3 respectively). The same difference was observed between the anomalous alpha exponent in CDC1 (pairwise $T$-test, $p = 1E$-5). These differences persisted when we fixed a constant number of 250 trajectories for all CDCs and conditions, indicating that these differences were not an artifact of change in data size (Supplementary Fig. 6C, D). Moreover, we did not observe any significant differences in the diffusion coefficient or the anomalous alpha exponent in scrambled CDCs at nuclear periphery or interior (Supplementary Fig. 6 E, F). Altogether, these results revealed faster diffusion of non-interacting particles in sparse regions at the nuclear periphery, which might be attributed to the effect of membrane pores[46] and which has interesting implications for the transport of nucleoplasmic proteins.

We also noted differences in nanoscale nucleosome organization between regions of similar chromatin density at the nuclear periphery and the nuclear interior. As anticipated, there is a noticeable enrichment of denser chromatin regions at the nuclear periphery. This is evident from the increased proportion of CDCs 5–7 at the periphery

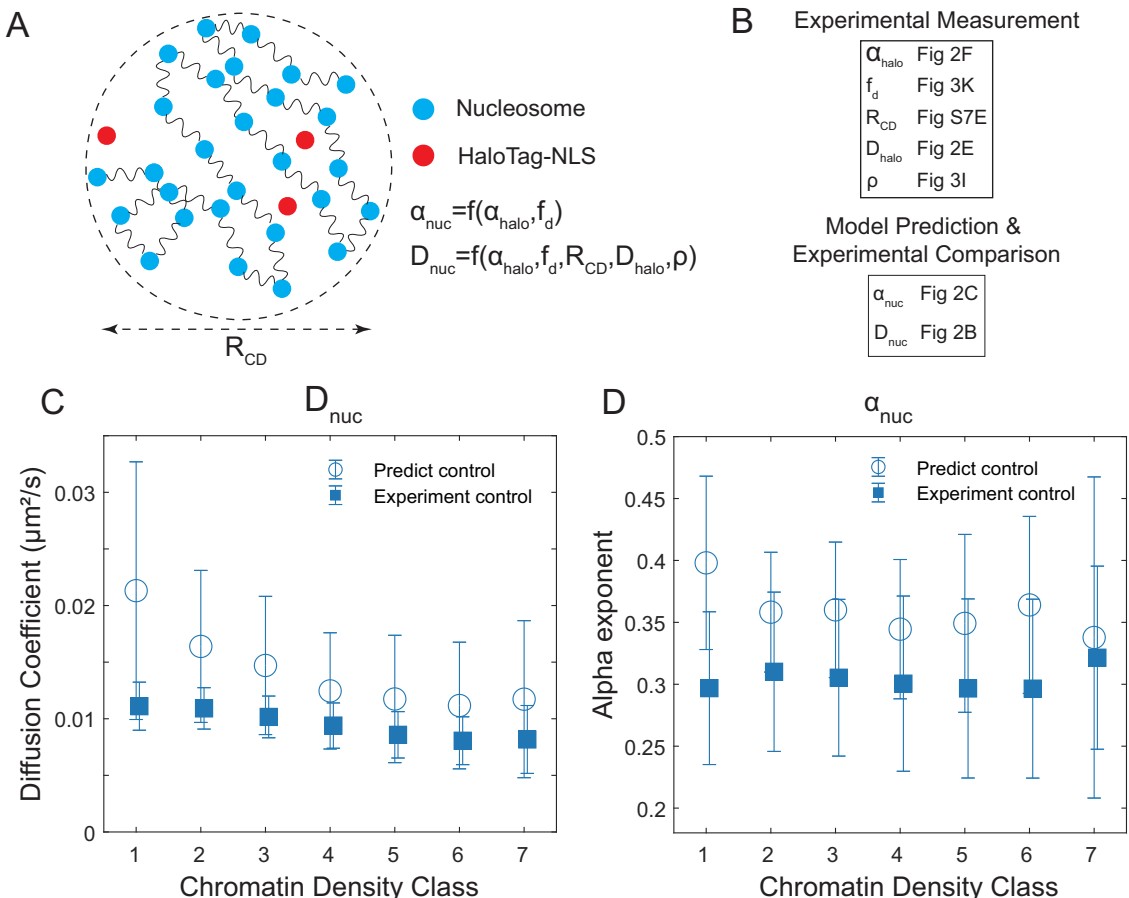

**Fig. 5 | A biophysical model predicts chromatin density class dependent nucleosome motion. A** Schematic of biophysical model of fractal polymer domain structure in a viscoelastic medium. Nucleosome anomalous exponent ($\alpha_{nuc}$) can be modeled as a function of the HaloTag-NLS anomalous exponent ($\alpha_{halo}$) and the estimated fractal dimension of the nucleosomes ($f_d$). The nucleosome diffusion coefficient ($D_{nuc}$) can be modeled as a function of the HaloTag-NLS anomalous exponent ($\alpha_{halo}$), the estimated fractal dimension of the nucleosomes ($f_d$), the size of a chromatin domain ($R_d$), the diffusion coefficient of the HaloTag-NLS ($D_{halo}$) and the nucleosome density ($\rho$). **B** Outline of empirical measurements and associated figures used to parameterize the biophysical model. **C** The comparison between the model-predicted and the experimentally measured (Fig. 2B) nucleosome diffusion coefficients across different chromatin density classes. Open circles indicate mean model predicted results and closed squares indicate mean experimentally measured results. The error bars for the experimental measured results represent the standard deviation of the experimental data. The error bars for the model predicted results are computed by propagating the error from the experimentally measured terms to the model output. **D** The comparison between model-predicted and experimentally measured (Fig. 2C) nucleosome anomalous alpha exponent across different chromatin density classes. The plot follows the same convention as (**C**). Experimental data from (**C**) and (**D**) are from $n = 88$ cells across 8 independent biological replicates.

compared to the interior (Supplementary Fig. 6G). However, when comparing individual CDCs at the nuclear periphery with those in the interior, we observed no significant differences in localization density (Supplementary Fig. 6H). Despite this, we did detect subtle changes in the estimated fractal dimension, with significant differences in in the sparser CDC1 and the intermediate CDCs 4–6 (Fig. 4J, Supplementary Fig. 6I, J). It's important to note that our estimates of fractal dimension in CDC1 might be less accurate due to the scarcity of CDC1 voxels at the nuclear periphery, and because this class comprises the fewest overall localizations. However, overall, these findings suggest that nucleosome organization at the nuclear periphery is distinctly influenced by elements of the nuclear lamina and is not directly comparable to regions of similar chromatin density in the nuclear interior. It also highlights the potential pitfalls of using nuclear periphery vs. interior as a proxy to study the specific effects of chromatin density.

### Biophysical models can integrate nucleosome motion, density, and packing properties

To integrate these results together with the nucleosome spatial organization and diffusive behavior observed above, we sought to

extend existing biophysical models of nucleosome motion to account for our analysis of chromatin density classes. Previous studies have modeled chromatin as a Rouse polymer comprised of nucleosomes that passively diffuse within a viscoelastic medium[47,48]. We specifically chose the model described in[48] because it incorporates the fractal organization of nucleosomes, the viscoelastic properties of the medium and has an analytical form for the predicted nucleosome MSD that can be compared to our experimental observations (Fig. 5A, B). As predicted in the model, the nucleosome MSD can be expressed as

$$\text{MSD}(t) = D_{nuc} t^{\alpha_{nuc}} \quad (1)$$

$$\alpha_{nuc} = \frac{2\alpha_{halo}}{2 + d_f} \quad (2)$$

$$D_{nuc} = C_{d_f, \alpha_{halo}} \left( \frac{3 D_{nucfree}}{\frac{4\pi}{3} \langle \rho \rangle_{CD}} \right)^{\frac{2}{2 + d_f}} \left\langle R_{CD}^2 \right\rangle^{\frac{d_f - 3}{d_f + 2}} \quad (3)$$

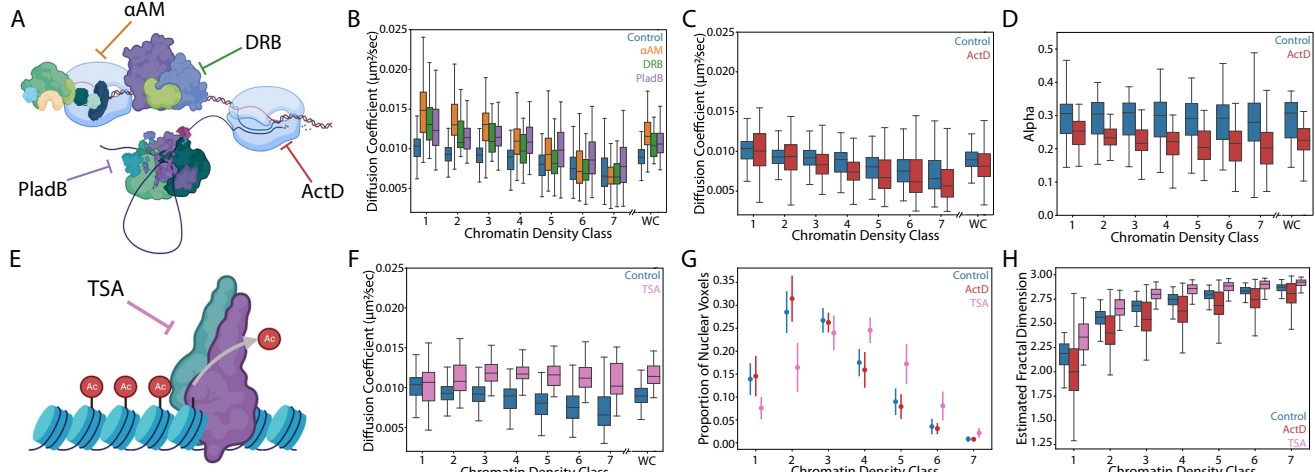

**Fig. 6 | Measurements of the relationship between nuclear functions and nucleosome properties. A** Schematic of different perturbations targeting gene transcription. **B** Box plot of diffusion coefficient in different chromatin density classes under control (blue), α-amanitin (orange), DRB (green) and PladB (purple). WC stands for whole cell. **C, D** Box plot of diffusion coefficient and anomalous alpha exponent in different chromatin density classes under control (blue), actinomycin D (red). **E** Schematic of Trichostatin A (TSA) perturbation on histone deacetylase activity. **F** Box plot of diffusion coefficient in different chromatin density classes under control (blue) and TSA (pink). **G** Point plot displaying the distribution of nuclear voxels in different chromatin classes under control (blue), ActD (red) and TSA (pink) conditions. The central dot indicates the mean proportion of voxels in given class for all cells. The vertical line indicates the standard deviation. **H** Box plot of estimated fractal dimension from super-resolution images of different chromatin classes under control (blue), ActD (red) and TSA (pink) conditions. Data from (**B**), (**C**), (**D**), (**F**) and (**G**) are from $n = 88$ cells across 8 replicates (control), $n = 52$ cells across 3 replicates (α-amanitin), $n = 46$ cells across 3 replicates (DRB), $n = 49$ cells across 3 replicates (PladB), $n = 41$ cells across 3 replicates (ActD) and $n = 60$ cells across 3 independent replicates (TSA). Data from (**H**) are from $n = 54$ cells (control), $n = 54$ cells (ActD), and $n = 60$ cells across 3 replicates (TSA). All replicates above were independent biological replicates. Box plots in (**B**–**D**), (**F**) and (**H**) follow same conventions as box plot in Fig. 2B.

In these equations, $C_{(d_f, \alpha\, halo)}$ is a dimensionless parameter determined from the chromatin fractal organization and the viscoelastic properties of the interchromatin space (see Methods). $\alpha_{halo}$ is the anomalous alpha exponent of a freely diffusing HaloTag, $D_{nuc\_free}$ is the apparent diffusion coefficient of a freely diffusing particle with a similar size as a nucleosome (for more details of $D_{nuc\_free}$ see Methods) within the interchromatin space, and $d_f$, $\langle \rho \rangle_{CD}$, and $\langle R_{CD}^2 \rangle$ are the fractal dimension, the nucleosome density, and the chromatin domain size respectively.

To compare this model to our observations, we estimated $\alpha_{halo}$ and $D_{nuc\_free}$ from our free-diffusing HaloTag-NLS measurements (Fig. 2E, F) and combined these with estimates of $d_f$, $\langle \rho \rangle_{CD}$, and $\langle R_{CD}^2 \rangle$ from our nucleosome measurements within each CDC ((Fig. 3K, Supplementary Fig. 7A–E) see Methods for details). We then compared the model-predicted nucleosome apparent diffusion coefficient and anomalous alpha exponent within each CDC to our experimentally measured values (Fig. 5C, D). We note that these comparisons arise from a model that is now fully parameterized with experimental data and has no free parameters. Because the nucleosomes' motion at nuclear periphery will be affected by their interaction with the lamina, which is not captured by this monodisperse polymer model, we only compared the model-predicted and the experimentally measured nucleosome motion in the nuclear interior. Overall, the model-predicted values for the nucleosome diffusion coefficient and the anomalous alpha exponent were in a similar range to our experimental measurements. The model also recapitulated a decrease in nucleosome diffusion coefficient with increased chromatin density (Fig. 5C) with only a modest dependence of the anomalous alpha exponent on chromatin density (Fig. 5D). Surprisingly though, the predicted sensitivity of the diffusion coefficient to chromatin density was much stronger than what we observed experimentally. At dense CDCs (CDC >= 4), the predicted nucleosome diffusion coefficient largely overlapped with our experimental measurements (predicted $0.0087 \pm 0.0037$ μm²/s vs. experimental $0.0077 \pm 0.0024$ μm²/s for CDCs 6-7), but the model substantially overestimated the nucleosome diffusion in sparse chromatin classes (predicted $0.0174 \pm 0.0068$ μm²/s vs. experimental $0.0099 \pm 0.0018$ μm²/s for CDCs 1–2).

Together, these results suggest that a fractal Rouse polymer model that incorporates the viscoelastic properties of the internuclear space can accurately predict nucleosome dynamics in dense chromatin regions, but that there appear to be additional factors that lead to slower than predicted nucleosome motion in sparse regions. This next led us to investigate what might be responsible for stabilizing nucleosomes within the low-density CDCs.

## Pharmacological perturbations alter nucleosome dynamics and suggest a partitioning of nuclear functions between chromatin density classes

Previous reports have found that proteins related to active processes, such as transcription and DNA looping, are enriched in areas of sparse and intermediate chromatin density[9]. For these reasons, we hypothesized that the discrepancy between model and experimental data could be due to active biological processes locally stabilizing nucleosome motion in sparse chromatin regions, as postulated in[49]. To test this hypothesis, we employed a panel of pharmacological perturbations to inhibit various steps of active transcription (Fig. 6A). Consistent with previous reports[10,18,19], our findings revealed that treatments including α-amanitin (αAM), 5,6-Dichloro-1-β-D-ribofuranosylbenzimidazole (DRB), and Pladienolide B (PladB) resulted in accelerated nucleosome dynamics throughout the entire nucleus (Supplementary Fig. 8A). Notably, the largest difference in nucleosome motion between the transcription inhibitors and control groups was observed in CDC1-3 (Fig. 6B). These differences decrease in magnitude through the intermediate classes, finally dropping below the level of significance in dense CDCs. The effects of these perturbations on nucleosome dynamics were primarily evident in the apparent diffusion coefficient, with no clear effects on the anomalous diffusion exponent (Supplementary Fig. 8B). Consistent with these results, in HCT116 cells in which the major sub-unit of RNA

polymerase II, RPB, has been engineered to be conditionally knocked down in the presence of the drug 5PH-I-AA[50] (Supplementary Fig. 8C), we see comparable, albeit less class specific, effects on nucleosome dynamics (Supplementary Fig. 8D, E).

Contrary to the effects observed with other pharmacological perturbations of transcription, but consistent with previous findings[18], we found that actinomycin D (ActD) treatment resulted in reduced nucleosome dynamics throughout the nucleus (Supplementary Fig. 8A), and across most CDCs (Fig. 6C). We also observed consistently more anomalous diffusion, as indicated by a decreased anomalous alpha exponent in all CDCs upon ActD treatment (Fig. 6D). Importantly, ActD operates through a distinct mechanism compared to the other tested perturbations. Unlike the other transcriptional perturbations, ActD does not act directly on the protein machinery but rather intercalates DNA, thereby hindering RNA polymerase from transcribing along the DNA template. Based on this mechanism, we propose that ActD induces a global stiffening effect on the DNA polymer. This hypothesis is supported by previous in vitro studies demonstrating the ability of ActD to alter DNA persistence length and contour length, ultimately resulting in increased stiffness[51].

To determine whether perturbations of other nuclear functions, beyond just transcription, might affect nucleosome motion in a chromatin-density dependent manner, we next investigated the effects of the broad-spectrum histone deacetylase inhibitor Trichostatin A (TSA) (Fig. 6E). Treatment with TSA leads to hyper-acetylation of the histone tails, ultimately weakening nucleosome-nucleosome contacts[52]. Previous studies have shown that TSA relaxes chromatin spatial organization[53,54] and increases average nucleosome dynamics across the nucleus[10]. Here, we show that TSA treatment increased nucleosome motion to the greatest extent in dense chromatin classes and abolished the relationship between nucleosome diffusion coefficient and CDC (one-way ANOVA F-stat = 1.45, $p$ = 0.19; Fig. 6F). TSA treatment did not induce changes anomalous alpha exponent (Supplementary Fig. 8F).

These data establish a compelling link between previous reports examining the relationship between chromatin density and specific markers of transcription in fixed cells and nucleosome dynamics in live cells and support the hypothesis that active transcription locally stabilizes nucleosomes in low-density CDCs[18]. To confirm this, we sought a more direct means of visualizing active transcriptional processes in live cells by monitoring the transcriptional bursting of endogenously regulated genes using the MS2 system in cells that were co-labeled with the DNA intercalating SiR-Hoechst dye (Supplementary Movie 5). We employed a previously established MCF7 cell line that stably expresses GFP-tagged MS2 (GFP-MCP) coat protein and contains 24x MS2 stem loops integrated into the 3' end of the endogenous loci of the estrogen response TFF1 gene[55]. When TFF1 is transcribed, GFP-MCP concentrates to the MS2 RNA stem loops, appearing as a bright green spot (Supplementary Fig. 9A). Consistent with our results suggesting that transcription primarily occurs in sparser chromatin regions, we found that the initiation of TFF1 transcription bursts primarily occurred in CDC1-2, with a smaller proportion in CDC3-4 (Supplementary Fig. 9B).

We found the relative impact of pharmacological treatments on nucleosome dynamics remains consistent when comparing the nuclear interior to nuclear periphery (Supplementary Fig. 10A–D). Furthermore, although the correlation between nucleosome diffusion coefficient and chromatin density dissipates at the nuclear periphery, the CDC specific effects of drug interventions on the nucleosome dynamics coefficient are largely maintained (Supplementary Figs. 11A, B, 12A, B). This observation suggests that, even with the disrupted link between nucleosome motion and chromatin density at the nuclear periphery, the functional compartmentalization within the nucleus is preserved.

## Pharmacological perturbations have differential effects on nucleosome dynamics and nucleosome organization

Unlike the distinct trends that link nucleosome dynamics with chromatin density classes upon transcription inhibition, inhibiting either the formation of the transcription pre-initiation complex using αAM or transcription elongation using DRB had no significant impact on nucleosome organization. This was evident at both the mesoscale, where no noticeable shift in the proportions of chromatin density classes was observed in our live-cell imaging (Supplementary Fig. 13A), and at the nanoscale, indicated by the consistent distribution of fractal dimensions across all classes similar to control cells (Supplementary Fig. 13B). Interestingly, we found that the spliceosome inhibitor PladB induced a small, though significant change in the proportional representation of CDCs in the nucleus in our live cell imaging (Supplementary Fig. 13A), as well as the apparent fractal dimension (Supplementary Fig. 13B). Previous studies indicate that ActD leads to increased chromatin compaction[56–58], whereas treatment with TSA is associated with chromatin relaxation[53,59]. While we did not observe significant changes in the proportional representation of CDCs following ActD treatment (Fig. 6G), we did find a more clustered nucleosome organization at the nanoscale. This was evidenced by a decrease in estimated fractal dimensions across all CDCs (Fig. 6H). Conversely, TSA appeared to have a 'normalizing' effect on chromatin organization at both the mesoscale and the nanoscale. At the mesoscale, TSA treatment resulted in a distribution of CDCs that was more Gaussian (Fig. 6G). At the nanoscale, TSA led to a more random organization of nucleosomes, as indicated by an increase in fractal dimensions across all CDCs (Fig. 6H). Consistent with the effects on nucleosome dynamics, the effects of all of the pharmacological treatments tested had similar effects on nuclear organization in the nuclear interior compared to the nuclear periphery (Supplementary Fig. 14A, B).

## Pharmacological perturbations do not alter viscoelastic properties of the interchromatin space with respect to small and inert proteins

It is possible that the observed effects of pharmacological perturbations on nucleosome dynamics might be explained by indirect changes to the viscoelastic properties of the interchromatin space (e.g. through decreased amounts RNA produced) rather than changes to chromatin itself. To test this, we measured the diffusive motion of non-interacting HaloTag-NLS under each of the pharmacological perturbations described above. Interestingly, although the perturbations did increase or decrease HaloTag-NLS motion across the nucleus as a whole when compared to control cells, none of the perturbations had a CDC-specific effect on HaloTag-NLS motion (Supplementary Fig. 15). An exception to this is ActD treatment which led to HaloTag-NLS exhibiting both slower and more anomalous diffusion across all CDC (Supplementary Fig. 15E, F). This result further supports hypothesis that ActD intercalation into DNA results in chromatin stiffening.

Finally, we tested how these perturbations to different nuclear functions affected the agreement between our experimentally observed results and the biophysical model described above (Fig. 7, Supplementary Fig. 16). Interestingly, transcription inhibitors αAM and DRB yielded better agreement between experimental results and model predictions for the nucleosome diffusion coefficients and the anomalous alpha exponents, especially in low-density CDCs. This suggests that gene transcription plays a role in actively stabilizing nucleosome motion within lower density CDCs in a manner that is distinct from the purely thermal forces considered in the model. We also found that TSA treatment led to the model performing worse in dense CDCs. This is likely because the model does account for electrostatic nucleosome-nucleosome interactions that are altered under TSA-induced hyper acetylation. These results highlight the need to

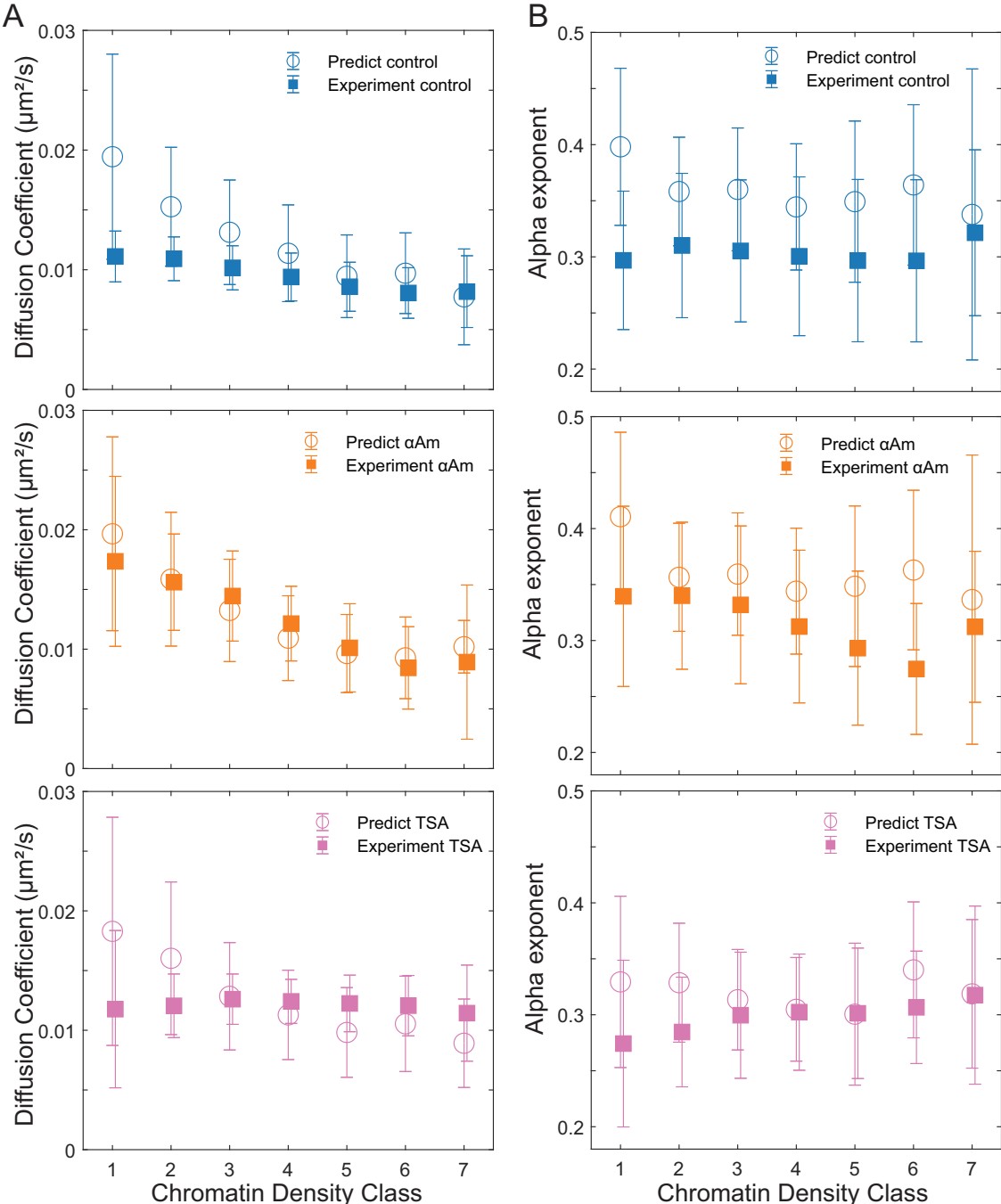

**Fig. 7 | Comparisons between model predicted and experimentally measured nucleosome dynamics under different pharmacological perturbations. A** The comparison between the model-predicted and the experimentally measured nucleosome diffusion coefficients across different chromatin density classes under control (blue), α-amanitin treatment (orange), and TSA treatment (pink). Plots follow same convention as Fig. 5C. **B** The comparison between model-predicted and experimentally measured nucleosome MSD exponent across different chromatin density classes under different conditions. The plot follows the same color assignments and convention as (**A**). Experimental data are from $n = 88$ cells across 8 replicates (control), $n = 52$ cells across 3 replicates (α-amanitin) and $n = 60$ cells across 3 independent replicates (TSA).

account for both active (non-thermal) forces as well as passive (thermal) forces when modeling chromatin dynamics.

## Discussion

A variety of biochemical reactions occur simultaneously within the nucleus of each cell. How these reactions are coordinated in absence of partitioning biological membranes is an active and ongoing area of investigation[60] that has been aided by technological developments in live cell[61], single molecule[10,16,18] and super resolution[8,9,53] imaging as well

as by theoretical modeling efforts in polymer sciences[14,48,62,63]. Here, we add to this field by developing an imaging approach to track individual nucleosomes and free diffusing proteins with nm precision while simultaneously reporting the local density and viscoelastic properties of the surrounding chromatin microenvironment at a resolution of a few hundred nanometers. With this methodology, we revealed that nucleosome motion is negatively correlated with local chromatin density. This finding agrees with and expands upon prior studies that explored nucleoosome motion by binarizing the nucleus into

heterochromatin vs euchromatin regions[16], but is distinct from reports of decreased nucleosome motion at the nuclear periphery[10]. While decreased nucleosome motion has previously been suggested to be due to increased heterochromatin at the nuclear periphery[10], it is important to note that the laminal region of the nucleus possesses additional distinct characteristics beyond chromatin density (reviewed in[64]). Specifically, the laminal DNA is physically tethered to the inner nuclear membrane[41,42], generally lacks active gene regulatory processes[44,45], and is enriched for distinct histone post translational modifications[44,65]. Here, we separately differentiate between the local chromatin density around a nucleosome and the nucleosome's distance from the nuclear periphery. Intriguingly, we observe that the anomalous alpha exponent of nucleosome motion is positively correlated with the distance from the nuclear periphery but is unaffected by local chromatin density. Furthermore, we find that nucleosome motion at the nuclear periphery is decoupled from local chromatin density. These findings suggest that there are likely additional effects, beyond changes to chromatin density, which impact nucleosome motion near the nuclear periphery. Compared with previous work that focused on either the dynamics[10,57] or the spatial organization[8,9,37,66] our approach provides an integrated view to investigate both properties in 3D simultaneously.

We complemented these measurements with fixed-cell, 3D super-resolution microscopy using spontaneously blinking Janelia-fluor probes[30] and found that nucleosome packing is more structured and clustered in sparse chromatin regions and becomes more optimally space filling and randomly organized in dense chromatin environments. Nucleosome packing displays a fractal-like organization between a length scale of ~40–200 nm, the lower-bound of which is imposed by our averaged 3D localization error. This length scale falls within the range of previously reported chromatin nanodomains[9,10,66], but is larger than the reported size of nucleosome clutches in which small groups of nucleosomes organize into discrete clusters[8,16]. This apparent discrepancy can be explained by our observation that the fractal dimension of nucleosome packing varies as a function of chromatin density, ranging from 2.1 in sparse CDCs to 2.8 in dense CDCs, a range which is comparable to that reported previously with variety of different methods[21,37,39,66]. A lower fractal dimension suggests that nucleosomes in sparser CDCs are more frequently organized into small, concentrated regions (such as clutches) separated by nucleosome free regions. In dense CDCs, as the inter-nucleosome spacing begins to approach the ~8 nm length scale of the nucleosome itself[67], the packing becomes more optimally space filling which results in a more random organization and a higher fractal dimension. These trends were also visually apparent from our super-resolution images (Supplementary Fig. 5).

Interestingly, the interchromatin space displayed predominantly viscous properties regardless of chromatin density, at least when probed by an inert HaloTag-NLS molecule. This indicates that from the perspective an inert nuclear protein of comparable size to a transcription factor, the nucleus is predominantly liquid-like and that this is independent of local chromatin density. This is consistent with earlier results measuring the viscoelastic properties of cell nucleus using quantum dots or fluorescent dextran[21,68] and indicates that even dense heterochromatin regions are still highly accessible. We want to emphasize that our delineations between chromatin dense and chromatin sparse CDCs only account for nucleosome density and do not report information about the concentrations of other nuclear proteins or nucleic acids. In fact, both chromatin-dense and chromatin-sparse CDCs can be highly crowded with other biomolecules. The fact that the dynamics of freely diffusing HaloTag-NLS show little dependence on local chromatin density suggest that the effective pore size for a diffusing nuclear protein of similar size to HaloTag-NLS is relatively homogeneous across the entire cell nucleus. The simple explanation for this is that regions with lower chromatin density are more crowded

with other biomolecules and visa-versa (Fig. 8) This is consistent with an evolving perspective on eu- and heterochromatin[69] as well as with prior studies that showed enrichment of other molecules in the interchromatin space and with measurements using orientation-independent-DIC microscopy that showed that heterochromatin is on average only 1.53 fold higher total density than euchromatin[56,70]. Although in this study, we did not specifically delineate between chromatin density classes and nucleolar associated regions, we anticipate that future applications of the approach could be extended to visualize single molecule diffusion in relation to a variety of nuclear landmarks including the nucleolus and nuclear speckles.

There have been several elegant attempts to connect polymer models[16,48,62,71] to experimentally measured nucleosome dynamics and spatial organization[16,39]. Our results complement these efforts by highlighting the heterogeneity in dynamics, organization, and biological function of chromatin and the challenges when treating the cell nucleus as a single material. For instance, a universal fractal globule model with a single fractal dimension may not fully describe the chromatin spatial organization[14]. Rather, we show that the apparent fractal dimension depends on the local chromatin density. The heterogeneity in chromatin dynamics and organization suggests similarities between the cell nucleus and colloidal glasses[72,73] and offers a possible explanation for how chromatin can actively undergo reorganization while maintaining its structural integrity. Moreover, biological processes such as gene transcription play a larger role in regulating nucleosome motion in chromatin sparse regions than chromatin dense regions. It will be important to include this heterogeneity, together with the role of active processes that may locally affect nucleosome dynamics in specific nuclear regions in future physical models of the cell nucleus. Of particular note, the model described here is fully parameterized by experimental data, containing no tunable parameters, which emphasizes its predictive accuracy and reliability.

Previous efforts to connect nucleosome dynamics to biological processes have produced conflicting results. Tracking single nucleosomes in RPE cells or single gene loci via an inserted MS2/TetR system revealed decreased chromatin motion during active gene transcription[18,74]. In contrast, tracking of several loci using dCAS targeted promoters and enhancers showed that chromatin motion was increased in settings wherein the genes were actively transcribing[75]. At the whole-nucleus level, our pharmacological perturbations agree with the model wherein transcription stabilizes nucleosome motion. Moreover, our finding that this stabilization predominantly occurs in low chromatin density regions, favors the model that transcriptional hubs with a higher local concentration of transcription regulatory machinery act to stabilize the local chromatin environment[18]. Our finding that this stabilization affects nucleosomes, but not the diffusion of free HaloTag-NLS molecules suggests several possible mechanisms. Chromatin stabilization could be accomplished through specific chemical interactions (e.g., between bromodomain containing proteins within the transcription hub to acetylated histone tails of the nucleosomes). Alternatively, it could arise from the generation of a local meshwork of RNA, DNA, and transcription machinery that has a pore size that is larger than single transcription factors, yet smaller than single nucleosomes. In this manner, a nuclear region could be stabilized in a transcriptionally competent state while allowing for free exchange of smaller and dynamic transcription factors and supporting machinery.

In summary, our work reveals the relationship between nucleosome dynamics, organization, nuclear viscoelasticity, nuclear location, and biological processes. Our findings support the need to account for physical and molecular heterogeneity in future biophysical models and will inform future studies of for how chromatin spatial organization and dynamics regulate cell development[76], disease[77,78], and stem cell reprogramming[8,16,79]. We envision that extensions of our imaging

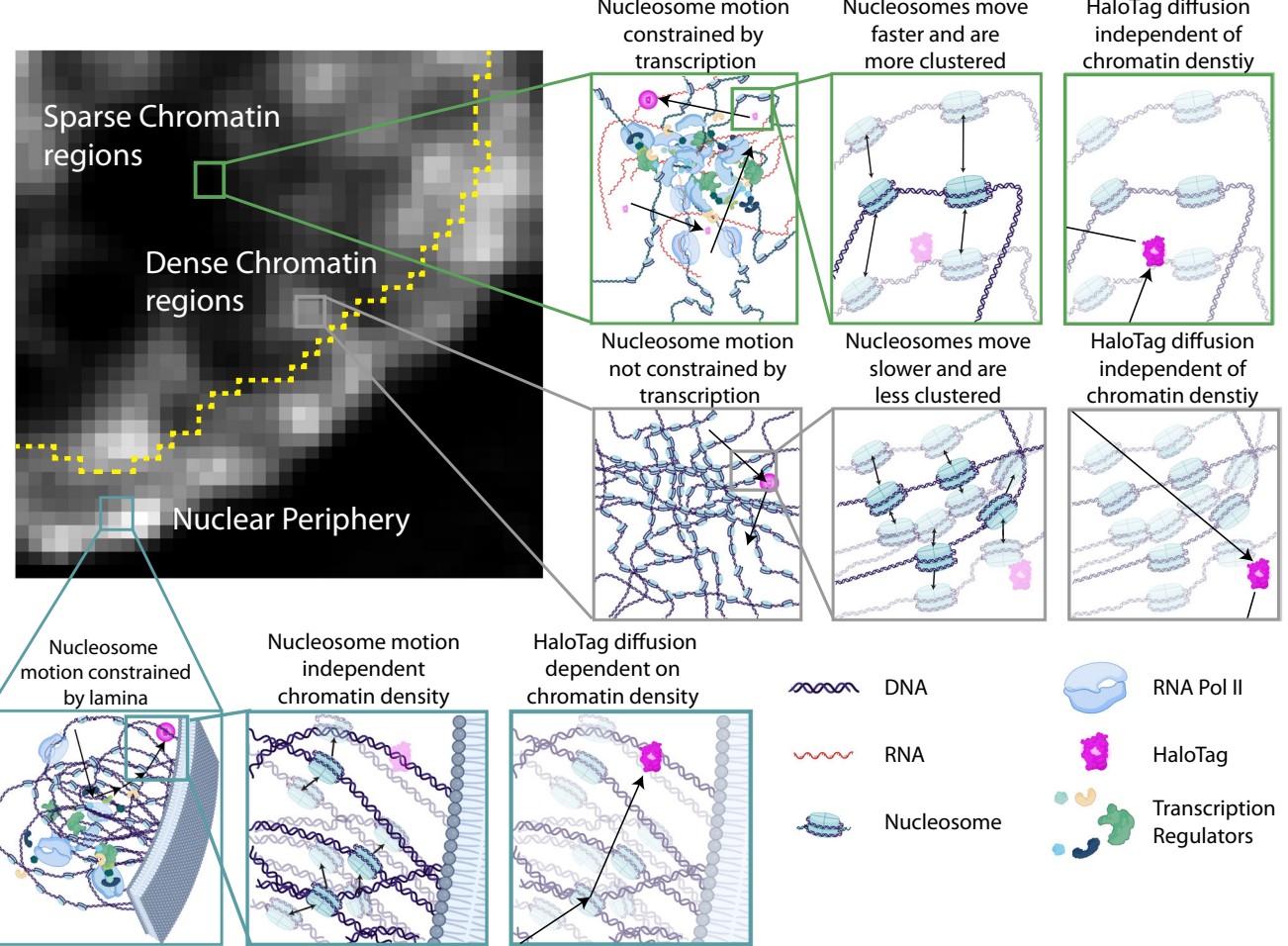

**Fig. 8 | Proposed model for chromatin density and organization.** Dark regions in the left panel represent chromatin sparse regions (green box). Nucleosomes in these regions form small clusters that are separated by nucleosome free regions and are more mobile than nucleosomes in denser chromatin regions. Their motion can be explained by a combination of passive diffusion and an active stabilizing component due to gene transcription. Bright regions in the left panel represent chromatin dense regions (gray box). Nucleosomes in these regions are more randomly organized and display less motion than those in chromatin sparse regions.

Their motion can predominantly be explained by passive diffusion. The diffusion of inert particles (magenta) in the nuclear interior is not affected by the local chromatin density (right column) or perturbations to transcription. Nucleosomes at the nuclear periphery (blue box) display distinct dynamics and organization from the nuclear interior. Nucleosome motion is more constrained but unaffected by local chromatin density. The diffusion of inert particles (magenta) at the nuclear periphery is affected by local chromatin density.

approach, in addition to other methods[80], to simultaneously visualize single molecules within the context of their local microenvironment will be useful in future studies to explore the relationship between chromatin density, nuclear function, and the diffusion of a diverse spectrum of nuclear proteins.

## Methods
### Sample preparation
**Cell culture.** Cos7-Halo-H2B and Cos7-Halo-NLS cells were maintained in DMEM (Thermo Fisher Scientific, 11965118), supplemented with 10% fetal bovine serum (Avantor 1300-500H) and 100 units/mL penicillin-streptomycin (Thermo Fisher Scientific, 15140122). HCT116 cells were maintained in McCoy's 5 A (Thermo Fisher Scientific 16600082) supplemented with 10% fetal bovine serum (Avantor 1300-500H) and 100 U/mL penicillin-streptomycin (Thermo Fisher Scientific, 15140122). MCF7- TFF1-MS2 cells were maintained in MEM (Thermo Fisher Scientific 11090081) supplemented with 10% fetal bovine serum (Avantor 1300-500H), 2 mM Glutamine (Thermo Fisher Scientific, 25030081) and 100 units/mL penicillin-streptomycin (Thermo Fisher Scientific, 15140122). Cells were routinely checked for mycoplasma contamination through both imaging and isothermal PCR methods.

**Generation of HaloTag-H2B and HaloTag-NLS cell lines.** Cos7-Halo-H2B cells were a generous gift of Luke Lavis. HCT116-mClover-mAID-RPB[50] cells were a generous gift of Dr. Masato Kanemaki at the National Institute of Genetics, Japan. Cos7-Halo-NLS and HCT116-mClover-mAID-RPB+Halo-H2B cell lines were generated using piggyBac transposon-based methods. The cDNA for HaloTag-NLS and HaloTag-H2B_BsdR genes was synthesized by GeneScript, then ligated into piggyBac plasmids that conferred resistance to either gentymycin or blasticidin[81,82]. Around $10^5$ cells were plated in 6 well plates and allowed to recover for 24 h. Cells were transfected with 1500 ng of HaloTag-NLS or HaloTag-H2B + 1000 ng of piggyBac Transposase (System Biosciences, PB210PA-1) using Lipofectamine2000 (Thermo Fisher Scientific, 11668027) and PLUS reagent (Thermo Fisher Scientific, 11514015) according to manufacturer's protocol. Two days post transfection, cells were harvested and plated into a 25 cm$^2$ flask in selection media. For Cos7-Halo-NLS cells, the selection media was DMEM (Thermo Fisher Scientific, 11965118), supplemented with 10% fetal bovine serum (Avantor 1300-500H) and 800 μg/mL of Geneticin (Thermo Fisher Scientific, 10131027). For HCT116-mClover-mAID-RPB +Halo-H2B cells, selection media was HCT116 growth media supplemented with 6 μg/mL of blasticidin (GoldBio, B-800-25).

**Co-localization of HaloTag-H2B and Hoechst sample preparation.** Cos7-Halo-H2B cells were plated on a 25 mm, #1.5 coverslips at a density of ~$10^4$ cell/cm$^2$ and allowed 24 h to recover from resuspension. Cells were incubated with 100 nM of HaloTag-Ligand JaneliaFluor 647 for 1 h, washed with pre-warmed PBS (Thermo Fisher Scientific, 10010023) then incubated in fresh growth media for 20 min.

4% paraformaldehyde solution was prepared through combining 10 mL of 16% PFA (Electron Microscopy Sciences 15710), 4 mL of 10x PBS (Corning 45001-130), and 26 mL of diH2O. Cells were washed briefly with PBS and fixed in room temperature 4% PFA for 12 min. Fixed cells were washed once with PBS for 5 s, followed by three 5-min washes in PBS (10x dilution of Corning 45001-130). All cell fixation and PBS wash steps followed this same protocol.

Fixed cells were incubated with 10 µg/mL of Hoechst (Thermo Fisher Scientific, H3570) for 20 min then washed according to previously described PBS washing scheme. Cells were mounted on 20 × 75 × 1.0 mm slide using vectashield (Vector Laboratories, H-1700-2) and sealed with nail polish 1 h later.

**Nuclear Extraction.** Cos7-Halo-H2B and Cos7 wild-type cells were grown to ~90% confluency in 100 mm plates. Cells were harvested by scraping in 4 °C PBS and centrifugation at 300x G for 3 min. The cell pellet was stored at −80 °C or immediately processed for nuclear extraction.

For nuclear protein isolation, the cell pellet was resuspended in 500 µL of nuclear extraction buffer containing 10 mM HEPES (Millipore-Sigma, H3375), 10 mM KCl (Millipore-Sigma, P3911), 0.5% Nonidet P-40 Substitute (Millipore-Sigma, 74385), 1 mM DTT (Millipore-Sigma, 11583786001), and 1x cOmplete Mini protease inhibitors (Millipore-Sigma, 11836153001). After a 10-min room temperature incubation, the sample was centrifuged at 13,000xG for 3 min to separate the cytoplasmic fraction from the nuclear fraction.

The nuclear pellet was then resuspended in 150 µL of lysis buffer containing 20 mM HEPES pH 8, 400 mM NaCl, 10% glycerol, 1 mM DTT (Millipore-Sigma, 11583786001), 1x cOmplete Mini protease inhibitors (Millipore-Sigma, 11836153001), and 1 µL/mL Pierce Universal Nuclease (Thermo Fisher Scientific, 88700), and incubated at 4 °C for 1H. A final centrifugation step at 10000x G for 5 min was performed to collect the supernatant containing the nuclear extract. The nuclear extract was then stored at −20 °C or used directly in Western blot analysis.

**Western Blot.** Proteins were separated by SDS-PAGE using a 10% polyacrylamide gel and transferred to a PVDF membrane. The membrane was blocked with 3% BSA in TBST for 1 h at room temperature. Blots were incubated with 500 ng/mL of anti-Histone H2B primary antibody (ABCAM, ab1790) or 1333 ng/mL of Histone H2B Monoclonal Antibody (Bioss, BSM-52099R, 3A6) overnight at 4 °C, followed by incubation with HRP-conjugated secondary antibodies (CytivaNA934-1ML) for 1 h at room temperature. Blots were imaged on a ChemiDoc MP Imaging system (BioRad, 12003154) and quantified using python (version 3.7, 3.8 or 3.9) (Supplementary Fig. 1B). Antibodies used in this study were confirmed by manufacturer via positive control Western blots against calf thymus histone preparation, and confirmed in our study by negative control western blots against wild type Cos7 cells that did not express HaloTag-H2B.

**Labeling of Halo-H2B cells for live cell imaging.** One day prior to imaging, cells were plated on 25 mm, #1.5 coverslips at a density of ~$10^4$ cells/cm$^2$. On the day of imaging, the cells were incubated in cell culture media containing 10 nM of the HaloTag-Ligand Photo-activatable JaneliaFluor 647 for 1 h. The cells were then washed with pre-warmed PBS (Thermo Fisher Scientific, 10010023) and incubated for an additional 20 min in growth media containing 100 nM of either HaloTag-Ligand JaneliaFluor 525 for Cos7-Halo-H2B cells or 100 nM of HaloTag-Ligand JaneliaFluor 549 for HCT116-mClover-mAID-RPB

+Halo-H2B cells. After the incubation, the cells were washed with pre-warmed PBS (Thermo Fisher Scientific, 10010023) and incubated in growth media containing no dye for 10 min.

**Labeling of Cos7-Halo-NLS cells for live cell imaging.** One day prior to imaging, cells were plated on 25 mm, #1.5 coverslips at a density of ~$10^4$ cells/cm$^2$. On the day of imaging, the cells were incubated in growth media containing 10 nM of the HaloTag-Ligand Photo-activatable JaneliaFluor 647 and a 500x dilution manufacturer recommended stock concentration of SPY505 (Cytoskeleton, CY-SC101) for 1 h. After the incubation, the cells were washed with pre-warmed PBS (Thermo Fisher Scientific, 10010023), and incubated in growth media containing no dye for 10 min.

**Labeling of cells for super resolution imaging.** Two days prior to imaging, cells were plated on 24 well #1.5 glass bottomed plates (Cellvis, P24-1.5H-N) at ~$10^5$ cells/cm$^2$. One day prior to imaging, cells were incubated in growth media supplemented with 500 nM HaloTag-Ligand Janelia Fluor 630b for 1 h. Cells were then washed with pre-warmed PBS (Thermo Fisher Scientific, 10010023) then incubated in fresh growth media for 20 min.

During the dye labeling steps, a 250 ng/mL solution of wheat germ agglutinin conjugated fluorescent nanodiamonds (Adámas, NDNV1000-WGA custom order) in PBS (10x dilution of Corning 45001-130) was sonicated for at least 1 h.

Cells were fixed and washed in PBS according to previously described methods (See Methods: Co-localization of HaloTag-H2B and Hoechst sample preparation). Cells were then incubated in the sonicated NDNV100nm-WGA solution for 1 h. Cells were washed in PBS according to previously described methods (See Methods: Co-localization of HaloTag-H2B and Hoechst sample preparation). Cells were imaged either immediately after the final wash step or stored at 4 °C for no >48 h.

**Pharmacological perturbations.** All small molecules used in this study meet community requirements for chemical probes. The following concentrations of drugs were used for pharmacological perturbations, α-Amanitin: 100 ug/mL (Millipore-Sigma A2263-1MG); 5,6-dichloro-1-beta-D-ribofuranosylbenzimidazole (Sigma-Aldrich D1916): 100 µM; Actinomycin D (Millipore-Sigma A9415-2MG): 0.5 µg/ml; Pladienolide B (Tocris, 6070) 30 ng/mL; Trichostatin A (Sigma-Aldrich T8552-5MG) 300 nM; 5PH-I-AA (MedChemExpress HY-134653): 1 µM. For pharmacological inhibition of transcription and RNA splicing with the drugs above, cells were incubated with the appropriate drug for 30 min prior to dye labeling. During dye labeling, the labeling and wash solutions were supplemented with the appropriate concentration of the drug. The total time of drug incubation for each of these conditions prior to imaging was 2 h. For pharmacological inhibition of histone deacetylases with Trichostatin A, cells were plated 2 days prior to imaging. Cells were incubated with Trichostatin A for 16 h prior to imaging. The dye labeling and wash solutions were supplemented with Trichostatin A. The total time of TSA incubation prior to imaging was 18 h. All live cell imaging was performed in Fluorobrite DMEM (Thermo Fisher Scientific A1896701), supplemented with 10% fetal bovine serum (Avantor 1300-500H), 2 mM Glutamine (Thermo Fisher Scientific, 25030081), 100 U/mL penicillin-streptomycin (Thermo Fisher Scientific, 15140122), and the drug concentrations listed above.

**MS2 sample prep.** TFF1-MS2 cells were described previously[55]. Transcription of the estrogen response gene was induced as previously reported. Cells were plated on 25 mm, #2 coverslips at least 3 days prior to imaging. At least 2 days prior to imaging, growth media was replaced with hormone depletion media, consisting of phenol free MEM (Thermo Fisher Scientific, 5120003), supplemented with 10% Charcoal stripped FBS (Millipore-Sigma F6765), 2 mM Glutamine

(Thermo Fisher Scientific, 25030081) and 100 units/mL penicillin-streptomycin (Thermo Fisher Scientific, 15140122). Media was replaced with fresh hormone depletion media once an hour, for a total of 3 h. Cells were then incubated in hormone depletion media for 24-72 h prior to induction. Prior to imaging, cells were incubated in fresh hormone depletion media supplemented with 1 μM Sir-Hoechst (Cytoskeleton, Cy-SC007) for 1 h. Cells were placed on the microscope in hormone depletion media supplemented with 1 nM Sir-Hoechst, and the XYZ position of 3-10 regions of interest were recorded. Immediately prior to beginning of time course imaging, 1 nM β-estradiol (Millipore-Sigma E8875-250MG) was added directly to imaging media.

## Imaging

**Confocal imaging of Cos7-Halo-H2B cells.** Confocal imaging was performed on a Zeiss LSM800 microscope using a 40x Oil immersion objective (NA 1.30) or 63x Water immersion objective (NA 1.20) using standard confocal imaging with a pinhole of 1 AU. Imaging was performed with separate channels for each wavelength, Alexa Fluor 647, Hoechst 33258, and T-PMT and a 2048×2048 pixel field of view.

**Optical setup for lattice light sheet and highly inclined swept tile.** The lattice light sheet imaging system was a modified version of the instrument described in[22]. Key modifications relevant to this work are the use of a greyscale spatial light modulator (Meadowlark P1920-0635-HDMI), a 0.6 NA excitation lens (Thorlabs, TL20X-MPL), and a 1.0 NA detection lens (Zeiss, Objective W Plan-Apochromat 20x/1.0, model #421452-9800). To achieve simultaneous multi-color imaging, we split the illumination light into offset strips on the spatial light modulator using a stack of dichroic elements (Semrock Di03-R405, Semrock Di03-R488, Semrock Di03-R561, Semrock MBP01), modulated each wavelength with the indicated lattice pattern on the spatial light modulator, and then recombined the reflected wavefronts by passing back through the same dichroic stack. For the emission path, we used a 642 nm super resolution dichroic (Semrock Di03-R635) to split the emission from JaneliaFluor 525 and JaneliaFluor 647 into two separate cameras (CamA and CamB respectively). To further filter the emission light and prevent channel bleed through from JaneliaFluor 525, a 514 nm long pass filter (Semrock BLP01-514R) and a 642 nm notch filter (Semrock NF03-642E) was placed in front of CamA. A 647 nm long pass filter (Semrock BLP01-647R) was placed in front of CamB. In addition, a 1000 mm focal length cylindrical lens (Thorlabs LJ516RM) was placed in front of CamB to generate astigmatism for 3D single molecule localization. To compensate for intensity changes in the chromatin channel due to photobleaching, we employed an adaptive power compensation based on the observed channel intensity at each time point. Briefly, after each image stack, we calculated the maximum intensity projection (projected through the z-axis). From this, we identified the 90[th] percentile pixel value and then linearly increased the laser power based on the change in this value from the previous time point.

Minor alterations in the imaging system were required for imaging of HCT116-mClover-mAID-RPB+Halo-H2B cells labeled with HaloTag-Ligand Janelia Fluor 549. A neutral density = 1.0 filter was placed in the 560 nm excitation path and a notch filter (Semrock NF03-561E) was placed in the emission path in front of CamA rather than 514 nm long pass filter. The optical path for CamB remained the same.

For single molecule localization microscopy (SMLM) we applied a highly inclined swept tile (HIST) microscope similar to the one described in ref. 31. To axially localize single molecules, a 1000 mm focal length cylindrical lens (Thorlabs LJ516RM) and Semrock FF01-680/42-32 emission filter was placed in front of a Hammamatsu BT Fusion camera.

**Live cell imaging procedure.** For single molecule tracking (SMT) imaging, we imaged each cell over 11 slices with a z-step size of 250 nm. At each slice, we collected 25 consecutive frames. This procedure was repeated for 50 times for each cell. Before and after imaging each cell, the dark current image of both cameras (no excitation was applied) was collected for 1000 frames to estimate the pixel-specific background of the image. For SMLM HIST imaging, we collected 100,000 frames for each field view to have enough number of localizations for reconstruction. The detailed imaging conditions for both cases are listed in Table 1.

## Image processing

**Pre- processing of live cell images.** For each cell, a single image for dark current subtraction was generated by taking the mean of the 2000 dark current images that were acquired before and after imaging each cell. The mean dark current image was then subtracted from all chromatin and SMT images. Any resulting negative voxel values were set to 0.

To adjust for minor chromatic offsets not corrected by initial microscope alignment, we periodically captured high signal-to-noise ratio volumetric images of chromatin in both cameras. We quantified the residual alignment discrepancy between the cameras down to the nearest voxel in all three dimensions (XYZ) using phase cross-correlation. Any detected offsets were then corrected in the dark current-corrected images prior to further image analysis.

**Chromatin images.** In order to minimize bleed-through of signal from the chromatin into simultaneously acquired SMT images, chromatin images were acquired under low signal to noise conditions. For each Z position, 25 dark current subtracted images were summed along the time axis and compiled into a high signal-to-noise z-stack. Compiled z-stacks were then deskewed to bring into a conventional XYZ reference frame and histogram equalized to the first full z-stack of each cell. To avoid edge artifacts in deconvolution, the deskewed Z-stacks were mirrored across the Z−axis 2 times. Mirrored images were then deconvolved using a Cuda implementation of the Richardson-Lucy algorithm using an experimentally measured lattice light sheet microscopy point spread function for 20 iterations (Supplementary Movie 6). The resulting deconvolved images were then cropped to match the dimensions of the original z-stack.

**SMT and SMLM image processing.** Prior to data acquisition, at least 15 z-stacks of 647 nm emission, 100 nm diameter beads were obtained. Super Resolution Microscopy Analysis Platform (SMAP)[83,84] was used

## Table 1 | Imaging conditions

| Condition | Figure | Illumination mode | Exposure | Laser λ, power (at objective pupil) |
|---|---|---|---|---|
| SMT | Figs. 1, 2, 4, and 6<br>Supplementary Figs. 1, 2, 3, 6, 8, 10, 11, 12, and 15 | LLS (MB-square, NA 0.4/0.3) | 20 ms | 405 nm, 26 μW<br>514 nm, 56 μW<br>642 nm, 9.75 mW |
| SMLM | Fig. 3<br>Supplementary Figs. 4, 5, 6, 7, 13, and 14 | HIST | 25 ms | 642, 40 mW<br>560 nm, 11.5 mW |
| MS2 | Supplementary Fig. 9 | LLS (MB-square, NA 0.4/0.3) | 10 ms | 642 nm, 25.7 μW<br>88, 27.3 μW |

to generate a 3D spline model of our astigmatic point spread function. SMT images were deskewed to bring them into conventional XYZ reference frame and cropped to match the same XYZ dimensions as the corresponding deconvolved images. Individual nucleosomes were then localized using SMAP. This was done by identifying regions of the image that contained individual nucleosomes using a difference of Gaussians filter followed by a local maximum filter. The nucleosomes within these regions were then localized in XYZ by fitting the 3D spline model.

From these localizations, trajectories were generated using a modified version of uTrack[85]. Our implementation of this algorithm set a maximum linking displacement of 400 nm, a maximum gap length of 2 frames, and disallowed merging and splitting of trajectories. The localizations were converted from nm to voxels through dividing the spatial coordinates of each localization by the voxel size and rounding to the nearest integer value. The voxel values for each localization were then used to determine the chromatin density class, and distance from the edge of the nucleus.

The procedure of HIST SMLM image processing is similar to SMT images. We first generated an astigmatic PSF model and applied it to extract the 3D localization coordinates using SMAP. We used the localization of nanodiamond fiducials to correct for spatial drift over the course of acquisition. Localizations that occurred within consecutive frames (with a gap of less than two frames) and that were found within a 100 nm radius were linked together into a single localization.

**Chromatin density classification of live cell images.** Briefly, binary images of the chromatin channel were generated using either an Otsu or a multi-Otsu thresholding method. Small holes and small binary objects in the mask were removed using a binary opening, followed by the skimage[86] functions remove_small_holes and remove_small_objects. Following this, all binary objects in the image were indexed using connected components. The largest of these labeled binary objects was kept for subsequent processing as a mask for each individual cell. The specific parameters for these operations, such as choice of thresholding method or maximum size of holes, were determined through visual inspection of the resulting masks. These masks were then applied to the deconvolved chromatin images prior to the assignment of nuclear voxels to different chromatin density classes using a previously described maximum likelihood estimation method using the bioimage tools and nucim R (version 4.102 or 4.2.2) libraries[27]. Briefly, this approach implements a random field Markov model, which takes into account the relative intensity of the voxel and the chromatin density class of neighboring voxels. After the chromatin density classification, to account for the roll off in intensity at the nuclear edge due to diffraction, all voxels in the least dense chromatin density class (CDC1) that were within 5 voxels of the edge of the nuclear mask were removed from the final chromatin density classification image (Fig. 1E).

**Scrambled chromatin density classification.** Deconvolved images were iteratively rotated and translated in X,Y, and Z. On every i-th iteration, a binary mask of the randomly transformed image was generated, and a binary mask of the i-1 iteration was also created. The voxels of the i-th iteration image that were not already occupied by non-zero values in the i-1 iteration's image were added to the i-1 image. These iterative transformations resulted in an object that was larger than the original deconvolved nucleus, composed of the randomly transformed gross morphological features of the original nucleus. After the iterative transformations, a binary mask image was generated by setting all non-zero voxels in the corresponding chromatin density image equal to one. This binary mask was applied to the randomly transformed image, and then the chromatin density classification was performed (Supplementary Fig. 3A, B).

**Distance from the nuclear periphery.** A binary mask was generated from the chromatin density class image by assigning a value of one to all non-zero voxels. A distance transform image was generated by calculating the distance of each non-zero voxel to the nearest zero-valued voxel (Fig. 4A). These distance transform images were used in both determination of the distance of single nucleosomes from the nuclear periphery as well as for voxel-wise comparison of CDCs and distance from the edge of the nucleus.

**Generation of simulated lattice light sheet microscopy images from SMLM localizations.** Lattice light sheet microscopy images from SMLM localizations were generated using Matlab (version 2021b, 2022a, 2022b or 2023a). Localizations were grouped into $110 \times 110 \times 110$ nm bins to construct a 3D histogram rendering. Simulated lattice light sheet microscopy images of the chromatin were subsequently generated by convolving this 3D histogram with an experimentally measured lattice light sheet microscopy point spread function obtained prior to live cell image acquisition. These simulated images were then deconvolved using parameters consistent with those applied in our live cell imaging (Supplementary Movie 4).

**Generation of the CDC images from simulated lattice light sheet microscopy images and classification of the SMLM localizations.** Generation of the CDC images from the simulated lattice light sheet microscopy images and classification of SMLM localization was performed using python (version 3.7, 3.8 or 3.9) and R (version 4.102 or 4.2.2). A preliminary binary mask was manually drawn around each cell within the simulated lattice light sheet microscopy image. Notably, any intensity regions associated with nanodiamond fiducials were excluded from these masks. For each cell, this mask was applied to the respective image. The highest intensity 110 nm z-plane was identified by summing the intensities along the X and Y axes. Any planes situated more than one z-plane above or below this plane were omitted from further analysis. The $\pm 1$ z-planes were then histogram matched to the central, brightest z-plane. The simulated single cell lattice light sheet microscopy images were then processed and classified into chromatin density classes using the methods previously described (See Methods: Chromatin density classification of live cell images).

**Partitioning the nuclear interior and nuclear periphery.** Masks of the nuclear interior and nuclear periphery were generated by binarizing the distance transform images (See Methods: Distance from Nuclear Periphery) according to a threshold distance of 500 nm from the outer edge of the nuclear mask. These masks were then applied to the CDC images and the scrambled CDC images.

**Data analysis**

**Statistical tests.** All statistical tests were performed using the SciPy statistical package for python (version 3.7, 3.8 or 3.9)[87]. Detailed results of these tests are available in the accompanying source data file.

**Analysis of HaloTag-H2B and Hoechst co-localization.** Images of HaloTag Janelia Fluorophre 647 and Hoechst were registered using phase cross correlation in order to correct for chromatic aberration. A Gaussian filter with a sigma value of 0.5 was applied to both images. Pixel-wise correlation was then performed using Pearson correlation (Supplementary Fig. 1A).

**Comparison of different number of chromatin density classes.** Deconvolved image stacks of chromatin images were masked according to the methods described above. The intensity values for masked, deconvolved images were then flattened into a one-dimensional array and all zero values were dropped. A mixture of Gaussian model was fit to the resulting intensity histograms with the number of classes ranging from 1 to 20. The Akaike information

criterion (AIC) and Bayesian information criterion (BIC) were determined for all fits. In order to compare the loss values across cells, the AIC and BIC for fits for all classes were z-score normalized (Supplementary Fig. 1D).

**Analysis of voxel-wise chromatin intensity and CDC dynamics.** A $1 \times 6$ voxel line profile was arbitrarily chosen in an example cell. The values for each voxel in the line profile were then recorded at all time points obtained (Supplementary Fig. 1E).

**Extracting diffusion coefficient and anomalous alpha exponent.** MSD was calculated as

$$\text{MSD}(\tau) = \left\langle \left(\mathbf{S}(t+\tau) - \mathbf{S}(t)\right)^2 \right\rangle_t \tag{4}$$

Here $\vec{s}(t)$ is the localization of a nucleosome at time t in 2D, $\tau$ is the time interval of the displacement, and $\langle X \rangle_t$ indicates the average over time. The squared displacements between all localizations within every trajectory were calculated at time intervals of up to 500 ms. The final localization of each displacement was used to assign it to a chromatin density class. MSDs were then averaged across all trajectories of the same class in the same cell, and a linear regression was applied to

$$\log(\text{MSD}) = \log(D) + \alpha * \log(\tau) \tag{5}$$

to extract the diffusion coefficient D and the anomalous alpha exponent (Fig. 1F). Extracting alpha anomalous alpha exponent and apparent diffusion coefficient of free diffusing HaloTag-NLS followed the same method as for nucleosomes except the squared displacements between all localizations within every trajectory were calculated at time intervals of up to 125 ms. Extracted parameters were only excluded from downstream analysis if the $r^2$ value of the linear fit was <0.95.

**Radius of gyration analysis.** For each trajectory in a given cell, all localizations were registered to the origin by subtracting the mean XY location of the trajectory from all localizations in said trajectory. The registered localizations were then segmented according to the CDC of the original localization. The radius of gyration was determined for each CDC by taking the root mean squared distance from the origin (Supplementary Fig. 2C).

**Angular anisotropy analysis.** The angle between every set of three subsequent localizations in a trajectory was determined. Said angles were then partitioned according to the CDC of the middle of the three points. Displacements were classified as 'forward' if the angle between displacements was between −30 and 30 degrees and 'backward' if the angle between displacements was between 150 and 210 degrees. For each CDC, the fold change between forward and backward displacements was determined by taking the $\log_2$ of the number of forward displacements divided by the number of backward displacements (Supplementary Fig. 2D).

**Power spectrum analysis of live cell and simulated lattice light sheet microscopy images.** For each cell, a square corresponding to a 5500 nm x 5500 nm in the center of the nucleus was converted from spatial domain to frequency domain using a Fast Fourier Transform. The power at each frequency was computed by calculating the squared magnitude of the Fourier coefficients.

A mesh grid representing different frequency components of the image was generated and the overall magnitude of frequency components in the 2D space was calculated. Radial frequencies were binned in bins starting at 0.5 and extending to half the pixel size in increments of 1. For each bin, the average of its boundaries was determined. The squared Fourier amplitudes were then grouped by their mean radial frequency and each binned amplitude was adjusted by multiplying it by the area of the corresponding annular region. The binned amplitudes were normalized by dividing by the maximum binned amplitude (Supplementary Fig. 4B).

**Proportion of voxels in CDCs.** For each cell, all CDC images were flattened into a 1D array and all zero values were dropped. From this, a normalized histogram was generated (Supplementary Fig. 4C).

**SMLM localization density.** For each cell, the number of SMLM localizations was determined for each CDC. The total volume of each CDC was calculated by counting the number of voxels in each CDC and then multiplying by the voxel dimensions in micrometers. Localization density was determined by dividing the total number of localizations in a CDC by its total volume (Fig. 3I).

**Visual comparison of SMLM localizations and random localizations in a given CDC.** The z-plane with the highest localization count was identified by scanning a 100 nm window in the z dimension and tallying the number of localizations. Localizations outside this z-plane were excluded from the super-resolution reconstructions. For each CDC, super-resolution reconstructions were generated by creating a 3D histogram using $20 \times 20 \times 100$ nm (XYZ) bins. Lines indicating the voxel boundaries of in the CDC image were drawn manually on the CDC image and then overlaid on the reconstructed SMLM image. For visual comparison, we generated randomly distributed localizations of the same density as our experimental measurements and plotted them within the same CDC boundaries (Supplementary Fig. 6).

**Pair correlation function calculation and fractal dimension estimation.** The normalized pair correlation function (G(r)) was calculated in a similar approach as described in[88], where

$$G(r) = \frac{N_R}{N} \frac{\text{DD}(r)}{\text{RR}(r)} \tag{6}$$

Here N is the total number of localizations in the actual data, and $N_R$ is the total number of localizations randomly distributed with the same density as the actual data. DD(r) is the average number of pairs of objects with separation r in the actual data, and RR(r) is the average number of pairs of objects with separation $r$ in the randomly distributed data. To account for diminished localization detection efficiency away from the focal plane, we adopted an approached similar to[37]. Briefly, for each cell, we plotted a histogram of the number of localizations found at each z-plane. We fit this histogram to a Gaussian function and then scaled the number of randomly distributed points in 3D space by this same Gaussian function. In this manner, the spatially random points were subject to the same axial sampling bias as our experimental measurements.

Based on the G(r) curve, we performed a linear regression on to log(G(r)) and log(r) to extract the decaying exponent $\gamma$ of G(r). The lower bound of the linear regression was set at 35 nm. The upper bound of the linear regression was determined through iteratively increasing the maximum value until the $r^2$ value of the fit fell below 0.98 ranging from the 35 nm to an upper value that was determined by a $r^2$ value of 0.98. The slope of this regression represents $\gamma$ and decays approximately as

$$G(r) \propto r^{-\gamma} \tag{7}$$

The fractal dimension is calculated as

$$d_f = 3 - \gamma \tag{8}$$

Extracted parameters were only excluded from downstream analysis if the $r^2$ value of the linear fit was >0.95. (Fig. 3K inset).

**Whole cell G(r) analysis.** For each $r$-value in our G(r) plot, spanning from 75 nm to 505 nm, we fit two linear regressions. The first regression used values from 75 nm to the $r$-value, while the second regression covered values from the $r$-value up to 505 nm. We then calculated the summed squared residual for both regressions across the respective ranges. The optimal hinge point was selected based on the $r$-value that produced the smallest combined summed squared residual. The estimated fractal dimension was derived from the slopes of these regressions, consistent with previously detailed methods (Supplementary Fig. 4D).

**Determination of nucleosome density.** Estimated nucleosome density is calculated as

$$\langle \rho \rangle_{CD} = \rho_{localization} * \frac{1}{N_{blinks}} \Big/ \frac{N_{h2b-halotag}}{N_{h2b}} \Big/ \frac{N_{cell:h2b-halotag}}{N_{cell}} \qquad (9)$$

where $\rho_{localization}$ is the density of localization. $N_{blinks}$ is the average number of blinks of each dye molecule as determined by the number of localizations in a DBSCAN cluster (Supplementary Fig. 7A, B).

$$\frac{N_{h2b-halotag}}{N_{h2b}} \qquad (10)$$

is the ratio of Halotag-H2B to endogenous h2b, which is determined by the Western blot (Supplementary Fig. 1B), and

$$\frac{N_{cell:h2b-halotag}}{N_{cell}} \qquad (11)$$

is the fraction of cells that express HaloTag-H2b (Supplementary Fig. 7C).

**Calculation of model predicted MSD characterizations.** Our overall procedure follows the formulas described in ref. 48. Briefly, the predicted anomalous alpha exponent for nucleosomes is calculated as

$$\alpha_{nuc} = \frac{2\alpha_{halo}}{2 + d_f} \qquad (12)$$

Where $\alpha_{halo}$ is the anomalous alpha exponent of free diffusing HaloTag-NLS and $d_f$ is the chromatin fractal dimension. The predicted nucleosome apparent diffusion coefficient is calculated as

$$D_{nuc} = C_{d_f,\alpha} \left( \frac{3 D_{nuc_{free}}}{\frac{4\pi}{3} \langle \rho \rangle_{CD}} \right)^{\frac{2}{2+d_f}} \langle R_{CD}^2 \rangle^{\frac{d_f-3}{d_f+2}} \qquad (13)$$

where $C_{d_f,\alpha}$ depends on $d_f$ and $\alpha$ and follows the expression

$$C_{d_f,\alpha} = \frac{2 B_{d_f,\alpha_{halo}}}{\left( A_{d_f} \right)^{\frac{d_f}{2+d_f}} \Gamma(1+\alpha_{halo})} \qquad (14)$$

where

$$A_{d_f} = \frac{\pi^{1+\frac{2}{d_f}}}{\Gamma\left(1+\frac{2}{d_f}\right) \sin\left(\frac{\pi}{d_f}\right)} \qquad (15)$$

and

$$B_{d_f,\alpha_{halo}} = \frac{d_f}{2} \left[ \Gamma(1+\alpha_{halo}) \right]^{\frac{d_f}{2+d_f}} \Gamma\left[\frac{d_f}{2+d_f}\right] \qquad (16)$$

This expression is based on previously published Eqs. 15 and 17 in ref. 48 and we substitute

$$\langle N \rangle_{CD} = \frac{4\pi}{3} \langle \rho \rangle_{CD} \langle R_{CD} \rangle^3 \qquad (17)$$

$D_{nuc\_free}$ is the diffusion coefficient of freely diffusing (non-chromatin bound) nucleosomes. This is different from the experimentally measured nucleosome diffusion coefficient as nucleosomes are connected by DNA. We estimated $D_{nuc\_free}$ by scaling the measured diffusion coefficient of HaloTag-NLS by the relative hydrodynamic radii of HaloTag-NLS and nucleosomes

$$D_{nuc_{free}} = D_{halo} \frac{R_{halo}}{R_{nuc}} \approx \frac{D_{halo}}{5} \qquad (18)$$

$\langle \rho \rangle_{CD}$ is the density of nucleosomes in chromatin domain, which we estimated based on the localization density in the super-resolution nucleosome data, and corrected for the blinking characterization of JF630B dye and the labeling efficiency of HaloTag-H2B (See Methods: Determination of nucleosome density). $\langle R_{CD}^2 \rangle$ is the averaged square size of the chromatin domain, and we estimated it based on the length where the slope of pair correlation function reaches zero (Supplementary Fig. 7D, E). The errors in the predicted anomalous exponent $\alpha_{nuc}$ and the predicted apparent diffusion coefficient of nucleosomes $D_{nuc}$ are estimated by propagating the errors in $\alpha$ and $d_f$ as estimated from experimental measurements.

## Reporting summary
Further information on research design is available in the Nature Portfolio Reporting Summary linked to this article.

## Data availability
Due to the inordinate size of the image data (~50TB), it is not currently feasible to deposit this into a central repository; however, all datasets underlying the results in this manuscript are available from the corresponding author upon request. To the extent possible, the authors will try to meet all requests for data sharing within 2 weeks from the original request. Source data are provided with this paper.

## Code availability
This study utilized established algorithms for image and statistical analysis as described in the Methods section, employing a combination of Matlab (version 2021b, 2022a, 2022b or 2023a), Python (version 3.7, 3.8 or 3.9) or R (Versions 4.1.2 or 4.2.2). There are no new or custom algorithms or software associated with this manuscript. Consequently, there is no unique code to be shared. Detailed descriptions of the methodologies can be found in the referenced sections.

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

## Acknowledgements

We thank John Crocker for helpful feedback and advice and Rachel Cherney, Mauro Calabrese, and Max Hockenberry for assistance with biochemical and imaging experiments. We also thank Victoria Augoustides for assistance in figure preparation. Figures 2A, D, 4F, H, J, 6A, E, 8, S6A, S6C, S6E, S6G, S8C were generated in part with BioRender.com. This work was funded in part by grants from the National Institutes of Health (1DP2GM136653) awarded to W.R.L. and (GM126900) awarded to B.D.S. W.R.L. acknowledges additional support from the Searle Scholars program, the Beckman Young Investigator Program, and the Packard Fellowship for Science and Engineering.

## Author contributions

W.R.L. conceived the project together with T.A.D. Y.S. and Z.L. L.D.L and K.L.H. generated the fluorescent Janelia-Fluor probes used in this study. H.R. J.R. and B.D.S. assisted with biochemical characterization of cell lines and protein expression quantification. T.A.D., Y.S., and W.R.L. performed the imaging experiments, analyzed the data, and wrote the

manuscript with feedback from all authors. W.R.L. supervised and directed the project.

## Competing interests

W.R.L. is an author on patents related to Lattice Light Sheet Microscopy and its applications including: U.S. Patent #'s: US 11,221,476 B2, and US 10,795,144 B2 issued to W.R.L. and coauthors and assigned to Howard Hughes Medical Institute. B.D.S is a co-founder of EpiCypher, Inc.T.A.D., Y.S., K.L.H, H.R., Z.L., L.D.L., and J.R. declare no competing interests.
