## [Peer Review File · Nature Communications]

Reviewers' Comments:

Reviewer #1:

Remarks to the Author:

How nucleosomes are organized as chromatin in the cell and how such organization is related to genome functions remain critical questions in cell biology. To approach these questions, Daugird et al. developed an imaging platform combining live-cell 3D single particle tracking (SPT) and high-resolution volumetric imaging via lattice light sheet microscopy. Using this state-of-the-art technology, the authors showed that, on average, nucleosomes in denser chromatin density classes (CDCs) display a slower apparent diffusion coefficient, suggesting a link between chromatin density and nucleosome dynamics. The authors also found that nucleosomes in low-density CDCs display a lower fractal dimension and more clustering. Those in high-density CDCs were more randomly distributed and more optimally space-filling. Furthermore, they investigated the effects of pharmacological perturbations on nucleosome dynamics and organization. The obtained findings are intriguing and advance our understanding of chromatin organization and functions in living cells. For publication in Nature Communications, my specific comments to improve this paper are as follows:

Major comments

1) Page 3, Line 102. I wonder how the authors obtained a high position determination accuracy (24 ± 9 nm (lateral) and 137 ± 59 nm (axial)) with the low magnification and NA of the detection objective (Page 21, Line 680 and Ref. 21). Please clarify this point.

2) Page 4 and Movie S2, Line 135. Related to Comment 1, how did the authors acquire the accurate z-position (137 ± 59 nm) from the tracked trajectories? Please clarify how to calibrate the z-position of single particle trajectories detected by u-track [75].

3) Page 4, Line 145. Why did the anomalous diffusion exponents display no significant differences across CDCs (Figure 1H), although the fractal dimension of nucleosome packing was positively correlated with chromatin density (Line 216)?

4) Page 4, Line 155. It would be intriguing to mention the nucleolar associated domains/regions (NADs), which are also heterochromatic.

5) Page 8, Line 236. HaloTag-NLS experiment is indeed informative. The data is well consistent with total density imaging of euchromatin and heterochromatin regions in mouse cells (PMID: 28835378).

6) Page 11 and Fig. 4, Line 323. I consider that TSA treatment can change the CDCs. In such a situation, are the CDCs still meaningful?

7) Hi-D maps (Ref. 51) claimed that DNA compaction and dynamics do not necessarily correlate, which seems to contradict the authors' conclusion. It would be useful to discuss this discrepancy.

Minor comments

1) Page 11 and Fig. 4, Line 323. For readers' better understanding, please briefly describe how TSA treatment changes chromatin: histone acetylation and weakened nucleosome-nucleosome contacts.

2) Page 11, Line 330. Why TSA treatment did not affect anomalous diffusion exponent?

3) A recent paper (PMID: 37385880) discussed how euchromatin and heterochromatin differ based on Refs. 9 and 14 and others. This issue is highly related to the main conclusion of this paper and would be better to be included.

Reviewer #2:

Remarks to the Author:

Daugrid and Shi et al. have developed a method enabling the simultaneous visualization of individual nucleosomes and chromatin density in living cells. Their methodology involved categorizing nuclear areas into seven distinct classes based on chromatin density, which enabled them to assess and quantify the diffusivity and packing properties of nucleosomes across these classifications. Key findings from their live-cell experiments indicate a trend where nucleosomes exhibit slower diffusion in regions with higher chromatin density. Contrastingly, in fixed-cell experiments, they observed a tendency for nucleosomes to cluster more densely in areas with less chromatin density, while more efficiently occupying space in regions of higher density, as revealed through fractal dimension analysis.

A significant aspect of their work involved measuring the diffusivity of the Halotag protein, leading to the discovery that the mobility of freely diffusing molecules remains unaffected by chromatin density. Utilizing this insight, alongside knowledge of the fractal organization of nucleosomes, the authors employed a polymer model to predict nucleosome diffusivity for each chromatin density class. While the model's predictions for higher density areas closely aligned with experimental results, the observed diffusivity in lower density areas was slower than predicted. This discrepancy led the authors to hypothesize that active transcription processes might be causing this reduced diffusivity. Subsequent experiments, involving global transcription inhibition, supported this hypothesis, showing a better alignment between predicted and observed diffusion constants in less dense chromatin areas.

This research corroborates previous findings and it also enhances our understanding through improved spatiotemporal resolution. The techniques introduced are a valuable addition to the field.

Comments:

1. Is it possible to analyze the dynamics of nucleosomes transitioning between different classes? Do dynamics of single nucleosomes change as they cross the threshold, or is the change more gradual? Information like this could provide further insight into the underlying chromatin structure. The authors should discuss/comment on this in the text if difficult or not possible.
2. Were the fractal dimensions of each chromatin density class measured post-drug treatment? It would be nice to determine if the fractal dimensions estimated from cells treated with drugs can predict nucleosome diffusivity after drugs. If not, measuring fractal dimensions post-drug treatment would be beneficial for validating the model's accuracy and understanding the drug's impact on chromatin structure.
3. The authors should provide further details/explanation as to why different microscopes were used for imaging in live versus fixed cells. This will help others with future methodological and/or experimental design.

Reviewer #3:

Remarks to the Author:

The manuscript entitled "Correlative single molecule lattice light sheet imaging reveals the dynamic relationship between nucleosomes and the local chromatin environment" is devoted to developing an imaging platform to simultaneously visualize single protein dynamics together with the local chromatin environment in live cells; and to better determine the relationship between chromatin density, nucleosome motion, the physical properties of the interchromatin space, and specific nuclear functions.

The authors claim in the conclusion (rows 490-497), that this proposed imaging method serves as a foundation "in future studies to explore the relationship between chromatin density, nuclear function, and the diffusion of a diverse spectrum of nuclear proteins".

The paper is very well written and structured in all sections. The topic would be a novel subject, to find NEWER biophysical properties and biological processes responsible for the observed differences in nucleosome motion, besides what is already known in this field. Unfortunately, the

authors have not stated in detail how their work adds to the work mentioned below, with arguments, as to why their findings are significant. The imaging approach using the lattice light-sheet system is new, but the authors goal to create a manuscript with newer findings has not been fulfilled. Therefore, the results don't represent a sufficiently striking advance (noteworthy results) for publication in Nature Communications. So, I would not recommend the manuscript for this journal. It could be a very good paper for cellular biochemistry or other cellular and molecular journals.

I will explain to the authors the major weaknesses of their manuscript, so that the authors can understand the basis for my decision.

1. I don't have comments on the originality, data analysis, interpretation, enough detail is provided in the methods for the work to be reproduced, as all these parts are clear.
2. The paragraph (rows 60-64) starts to build the storyline of this paper, using two manuscripts (defined by the authors as 'recent work'), one from 2022 (ref # 19) and the other one from 2009 (ref. #20), ignoring the fact that there are many other findings in these topics between the two references. Because the results/findings of the authors replicate already published existing results, the results of this manuscript are very observational for a Nature journal.
3. Based on the statement below, the authors claim that " We hypothesized that the observed differences in nucleosome motion across CDCs may arise from various factors, including variations in nucleosome spatial packing, differences in the local viscoelastic properties of the material surrounding chromatin, and/or differences in active processes that may be localized to a given CDC. These factors, either individually or in combination, could contribute to the distinct patterns of nucleosome dynamics observed in different CDCs. Subsequently, our investigation aimed to identify and characterize the biophysical properties and biological processes responsible for these observed differences in nucleosome motion."

These hypotheses and findings were already covered in various published work, as follows:

- A) chromatin exhibits a polymeric fractal-like behavior and a radially decreasing mass-density from the center to the periphery.
- B) for the majority of the cells the Dlog at the periphery was again higher than Dlog at the non-periphery

Li, Y., Agrawal, V., Virk, R.K.A. et al. Analysis of three-dimensional chromatin packing domains by chromatin scanning transmission electron microscopy (ChromSTEM). *Sci Rep* 12, 12198 (2022). <https://doi.org/10.1038/s41598-022-16028-2>

- C) "more compact chromatin located remote from IC channels can be attributed to the INC. It is enriched in epigenetic marks for low or silent transcriptional activity"

Popken J, Brero A, Koehler D, Schmid VJ, Strauss A, Wuensch A, Guengoer T, Graf A, Krebs S, Blum H, Zakhartchenko V, Wolf E, Cremer T. Reprogramming of fibroblast nuclei in cloned bovine embryos involves major structural remodeling with both striking similarities and differences to nuclear phenotypes of in vitro fertilized embryos. *Nucleus*. 2014;5(6):555-89. doi: 10.4161/19491034.2014.979712. PMID: 25482066; PMCID: PMC4615760.

Cremer T, Cremer M, Hübner B, Silahtaroglu A, Hendzel M, Lanctôt C, Strickfaden H, Cremer C. The Interchromatin Compartment Participates in the Structural and Functional Organization of the Cell Nucleus. *Bioessays*. 2020 Feb;42(2):e1900132. doi: 10.1002/bies.201900132. PMID: 31994771.

- D) Figure 6, the same conclusion as:

Márton Gelléri, Shih-Ya Chen, Barbara Hübner, Jan Neumann, Ole Kröger, Filip Sadlo, Jorg Imhoff, Michael J. Hendzel, Marion Cremer, Thomas Cremer, Hilmar Strickfaden, Christoph Cremer, "True-to-scale DNA-density maps correlate with major accessibility differences between active and inactive chromatin", *Cell Reports*, Volume 42, Issue 6, 2023, 112567, ISSN 2211-1247, <https://doi.org/10.1016/j.celrep.2023.112567>.

Shinkai S, Nozaki T, Maeshima K, Togashi Y (2016) Dynamic Nucleosome Movement Provides Structural Information of Topological Chromatin Domains in Living Human Cells. *PLOS Computational Biology* 12(10): e1005136. <https://doi.org/10.1371/journal.pcbi.1005136>

Bizhanova A, Kaufman PD. Close to the edge: Heterochromatin at the nucleolar and nuclear peripheries. *Biochim Biophys Acta Gene Regul Mech.* 2021 Jan;1864(1):194666. doi: 10.1016/j.bbagr.2020.194666. Epub 2020 Dec 8. PMID: 33307247; PMCID: PMC7855492.

Bajpai G, Amiad Pavlov D, Lorber D, Volk T, Safran S. Mesoscale phase separation of chromatin in the nucleus. *Elife.* 2021 May 4;10:e63976. doi: 10.7554/eLife.63976. PMID: 33942717; PMCID: PMC8139833.

E) "fractal dimension depends on the local chromatin density"

Almassalha, L., Tiwari, A., Ruhoff, P. et al. The Global Relationship between Chromatin Physical Topology, Fractal Structure, and Gene Expression. *Sci Rep* 7, 41061 (2017). <https://doi.org/10.1038/srep41061>

Récamiér V, Izeddin I, Bosanac L, Dahan M, Proux F, Darzacq X. Single cell correlation fractal dimension of chromatin: a framework to interpret 3D single molecule super-resolution. *Nucleus.* 2014 Jan-Feb;5(1):75-84. doi: 10.4161/nucl.28227. Epub 2014 Feb 19. PMID: 24637833; PMCID: PMC4028358.

F) "there are a variety of techniques to measure chromatin accessibility and related quantities using chemical probes"

Mansidor AR, Risca VI. Chromatin accessibility: methods, mechanisms, and biological insights. *Nucleus.* 2022 Dec;13(1):236-276. doi: 10.1080/19491034.2022.2143106. PMID: 36404679; PMCID: PMC9683059.

Tsompana, M., Buck, M.J. Chromatin accessibility: a window into the genome. *Epigenetics & Chromatin* 7, 33 (2014). <https://doi.org/10.1186/1756-8935-7-33>

G) "biological processes such as gene transcription play a larger role in regulating nucleosome motion in chromatin sparse regions than chromatin dense regions"

Michael Babokhov, Kayo Hibino, Yuji Itoh, Kazuhiro Maeshima, "Local Chromatin Motion and Transcription", *Journal of Molecular Biology*, Volume 432, Issue 3, 2020, Pages 694-700, ISSN 0022-2836, <https://doi.org/10.1016/j.jmb.2019.10.018>.

H) "transcription stabilizes nucleosome motion, but in some cases could destabilize the motion, it depends on the specific position and orientation of the binding motifs relative to the nucleosome"

Ekaterina Morgunova, Jussi Taipale, "Structural insights into the interaction between transcription factors and the nucleosome", *Current Opinion in Structural Biology*, Volume 71, 2021, Pages 171-179, ISSN 0959-440X, <https://doi.org/10.1016/j.sbi.2021.06.016>.

Lai WKM, Pugh BF. Understanding nucleosome dynamics and their links to gene expression and DNA replication. *Nat Rev Mol Cell Biol.* 2017 Sep;18(9):548-562. doi: 10.1038/nrm.2017.47. Epub 2017 May 24. PMID: 28537572; PMCID: PMC5831138.

I) "the diffusion of free molecules increases in chromatin less dense regions"

Mazzocca, M., Loffreda, A., Colombo, E. et al. Chromatin organization drives the search mechanism of nuclear factors. *Nat Commun* 14, 6433 (2023). <https://doi.org/10.1038/s41467-023-42133-5>

Conclusion:

Known facts:

- the nucleosome motion is negatively correlated with local chromatin density
- the decreased nucleosome motion is due to increased heterochromatin at the nuclear peripher, forming Lamina-associated domains

- the fractal dimension of nucleosome packing was also positively correlated with chromatin density
- the nucleosome packing is more structured and clustered in sparse chromatin regions and becomes more optimally space filling and randomly organized in dense chromatin environments
- the diffusion of free molecules increases in chromatin less dense regions

NCOMMS-23-56690 Author Response to Reviewer Comments

Below, we summarize the reviewer's feedback on the original manuscript and provide detailed responses to each of their comments. We thank the editor and reviewers for their time and feedback on our original manuscript and believe that the revised submission is much improved as a result of this process. To aid readability, major changes in the manuscript revision are indicated in **dark red font**. Reviewer comments below are in black font and our responses are in **blue font**.

REVIEWER COMMENTS

Reviewer #1 (Remarks to the Author):

How nucleosomes are organized as chromatin in the cell and how such organization is related to genome functions remain critical questions in cell biology. To approach these questions, Daugird et al. developed an imaging platform combining live-cell 3D single particle tracking (SPT) and high-resolution volumetric imaging via lattice light sheet microscopy. Using this state-of-the-art technology, the authors showed that, on average, nucleosomes in denser chromatin density classes (CDCs) display a slower apparent diffusion coefficient, suggesting a link between chromatin density and nucleosome dynamics. The authors also found that nucleosomes in low-density CDCs display a lower fractal dimension and more clustering. Those in high-density CDCs were more randomly distributed and more optimally space-filling. Furthermore, they investigated the effects of pharmacological perturbations on nucleosome dynamics and organization. The obtained findings are intriguing and advance our understanding of chromatin organization and functions in living cells. For publication in Nature Communications, my specific comments to improve this paper are as follows:

Major comments

1) Page 3, Line 102. I wonder how the authors obtained a high position determination accuracy (24 ± 9 nm (lateral) and 137 ± 59 nm (axial)) with the low magnification and NA of the detection objective (Page 21, Line 680 and Ref. 21). Please clarify this point.

We apologize for the confusion. These localization statistics refer to single particle tracking performed via lattice light sheet microscopy. Although the instrument utilizes a "20X" detection objective, the final magnification is 60X which is sufficient to Nyquist sample the 1.0 numerical aperture of the lens with the 6.5 micron pixel size of the Orca Flash 4.0 camera. The localization precision is determined from the Cramer Rao lower bound of the fit as computed in the SMAP fitting software package. Although 1.0 NA is relatively low compared to other objective lenses used in localization microscopy, the very low background attained by lattice light sheet illumination allows for high-precision fits. We note that these precision estimates are comparable to those presented previously in (Liu et al. Elife 2014 and Legant et al. Nature Methods 2016).

2) Page 4 and Movie S2, Line 135. Related to Comment 1, how did the authors acquire the accurate z-position (137 ± 59 nm) from the tracked trajectories? Please clarify how to calibrate the z-position of single particle trajectories detected by u-track [75].

We note in the Methods section under the heading "SPT and SMLM image processing" that single nucleosome localizations are localized through fitting a 3D spline model. Our original submission cited: J. Ries, SMAP: a modular super-resolution microscopy analysis platform for SMLM data, Nat. Methods 17 (2020) 870–872. <https://doi.org/10.1038/s41592-020-0938-1>

as the appropriate reference for both calibrating and fitting such models. In this revision, we have added an additional reference:

Y. Li, M. Mund, P. Hoess, J. Deschamps, U. Matti, B. Nijmeijer, V.J. Sabinina, J. Ellenberg, I. Schoen, J. Ries, Optimal 3D single-molecule localization in real time using experimental point spread functions, *Nat. Methods* 15 (2018) 367–369. <https://doi.org/10.1038/nmeth.4661>.

to further clarify the methods that were used for fitting single nucleosomes in XYZ.

After localizing nucleosomes in XYZ with SMAP, u-track was utilized for linking localizations across frames to generate trajectories.

3) Page 4, Line 145. Why did the anomalous diffusion exponents display no significant differences across CDCs (Figure 1H), although the fractal dimension of nucleosome packing was positively correlated with chromatin density (Line 216)?

We thank the reviewer for raising this point. We speculate this is because both the chromatin structure as well as other molecules in the nucleoplasm and the viscoelasticity of nucleoplasm itself will affect the anomalous diffusion exponent of nucleosomes. As reported in Fig 3, we find little variation in the anomalous diffusion exponent of non-interacting halo-tag, which reflects that the overall viscoelasticity of cell nucleus shows little variation across different chromatin density.

4) Page 4, Line 155. It would be intriguing to mention the nucleolar associated domains/regions (NADs), which are also heterochromatic.

We thank the reviewer for this suggestion. To acknowledge this point, we have added a passage in the discussion stating that one of the limitations of our study is the inability to differentiate between chromatin density effects and those arising from other nuclear landmarks such as speckles or the nucleolus. We further highlight that the potential to extend our imaging platform to explore these aspects, including NADs and potentially other nuclear landmarks presents an exciting avenue for future research. Specifically, we state:

“Although in this study, we did not specifically delineate between chromatin density classes and nucleolar associated regions, we anticipate that future applications of the approach could be extended to visualize single-molecule diffusion in relation to a variety of nuclear landmarks including the nucleolus and nuclear speckles.”

5) Page 8, Line 236. HaloTag-NLS experiment is indeed informative. The data is well consistent with total density imaging of euchromatin and heterochromatin regions in mouse cells (PMID: 28835378).

Thank you for drawing our attention to this reference. We have ensured that this paper is included in references during our discussion of dynamics of freely diffusing HaloTag in sparse and dense chromatin regions.

6) Page 11 and Fig. 4, Line 323. I consider that TSA treatment can change the CDCs. In such a situation, are the CDCs still meaningful?

We appreciate the reviewer's insightful comment and careful reading of our manuscript. We acknowledge that TSA treatment influences mesoscale chromatin organization, as reflected in the altered distribution of CDCs. It is important to clarify that the designation of CDCs in our study is a statistical tool for classifying the smooth distribution of H2b intensities into discrete categories. This approach allows for the comparison of subnuclear regions based on similar local densities of chromatin,

rather than delineating physically distinct nuclear territories (e.g. nanodomains, clutches etc). We applied this statistical partitioning consistently to both control and treatment groups and therefore believe that this method remains an unbiased way to compare changes in local chromatin density across different conditions.

7) Hi-D maps (Ref. 51) claimed that DNA compaction and dynamics do not necessarily correlate, which seems to contradict the authors' conclusion. It would be useful to discuss this discrepancy.

Thank you for this comment. Hi-D is a powerful method for bulk chromatin dynamics in live cells, however, as the Authors of the Hi-D reference above note:

“One should, however, keep in mind that SPT and Hi-D are meant to analyze images from drastically different labeling conditions and should thus refrain from a direct comparison between single-locus dynamics analyzed by SPT and local bulk chromatin dynamics by Hi-D”

In the supplement of this paper, Shaban and co-authors discuss the methodological and physical factors leading to discrepancies between SPT and HiD data, emphasizing the inherent difficulty in comparing these methods under realistic imaging conditions.

Despite this, one might still expect more general observations to be preserved across orthogonal methodologies. We propose two potential additional factors to explain the observed discrepancies:

1. The disparity in the physical scale and composition of the objects measured: SPT tracks the movement of a single nucleosome typically over 10^0 - 10^2 ms, whereas HiD assesses the motion of larger chromatin structures composed of thousands of nucleosomes, generally over 10^2 - 10^3 ms. It remains possible that the influence of local chromatin environment on nucleosome motion may not be directly scalable to the larger structures observed in HiD.
2. Differing levels of sensitivity: The variation in diffusion coefficients we observed across CDCs is about $0.004 \mu\text{m}^2/\text{sec}$, ranging from sparse to dense chromatin regions. This level of variation might be below the detection precision of methods like HiD.

Minor comments

1) Page 11 and Fig. 4, Line 323. For readers' better understanding, please briefly describe how TSA treatment changes chromatin: histone acetylation and weakened nucleosome-nucleosome contacts.

We thank the reviewer for this suggestion and incorporated your suggestion and supporting reference in the revised manuscript.

2) Page 11, Line 330. Why TSA treatment did not affect anomalous diffusion exponent?

We speculate this is because the anomalous diffusion coefficient is impacted by the viscoelasticity of the environment that nucleosomes interact with. This will include not only chromatin, but nucleoplasm and other proteins in cell nuclei. Therefore, although TSA treatment increases the fractal dimension of chromatin, it can also increase the binding of trans-regulatory factors that could stabilize chromatin, and possibly compensate the effect of its modification on the chromatin structure. In fact, as shown in Fig S9 of the original submission, the overall anomalous diffusion coefficient measured by non-interacting halo-tag show little variation under TSA treatment, indicating that the overall viscoelasticity of nucleus was unchanged.

3) A recent paper (PMID: 37385880) discussed how euchromatin and heterochromatin differ based on

Refs. 9 and 14 and others. This issue is highly related to the main conclusion of this paper and would be better to be included.

Thank you for bringing this to our attention. We agree that our understanding of eu- and heterochromatin are constantly evolving challenging previously considered concepts. We have included this manuscript as a reference in the revised submission.

Reviewer #2 (Remarks to the Author):

Daugrid and Shi et al. have developed a method enabling the simultaneous visualization of individual nucleosomes and chromatin density in living cells. Their methodology involved categorizing nuclear areas into seven distinct classes based on chromatin density, which enabled them to assess and quantify the diffusivity and packing properties of nucleosomes across these classifications. Key findings from their live-cell experiments indicate a trend where nucleosomes exhibit slower diffusion in regions with higher chromatin density. Contrastingly, in fixed-cell experiments, they observed a tendency for nucleosomes to cluster more densely in areas with less chromatin density, while more efficiently occupying space in regions of higher density, as revealed through fractal dimension analysis.

A significant aspect of their work involved measuring the diffusivity of the Halotag protein, leading to the discovery that the mobility of freely diffusing molecules remains unaffected by chromatin density. Utilizing this insight, alongside knowledge of the fractal organization of nucleosomes, the authors employed a polymer model to predict nucleosome diffusivity for each chromatin density class. While the model's predictions for higher density areas closely aligned with experimental results, the observed diffusivity in lower density areas was slower than predicted. This discrepancy led the authors to hypothesize that active transcription processes might be causing this reduced diffusivity. Subsequent experiments, involving global transcription inhibition, supported this hypothesis, showing a better alignment between predicted and observed diffusion constants in less dense chromatin areas.

This research corroborates previous findings and it also enhances our understanding through improved spatiotemporal resolution. The techniques introduced are a valuable addition to the field.

Thank you for highlighting that we both corroborate previous findings and also enhance our understanding through new results/analysis.

Comments:

1. Is it possible to analyze the dynamics of nucleosomes transitioning between different classes? Do dynamics of single nucleosomes change as they cross the threshold, or is the change more gradual? Information like this could provide further insight into the underlying chromatin structure. The authors should discuss/comment on this in the text if difficult or not possible.

We appreciate the reviewer's suggestion. As we discuss in the results section under the "Lattice light sheet microscopy enables simultaneous 3D tracking of individual nucleosomes together with high-resolution measurement of local chromatin density" heading, Chromatin Density Classes (CDCs) are a statistical partitioning of a continuous intensity distribution, not indicative of physically distinct chromatin regions (as has been described previously as nanodomains and clutches). Consequently, our analysis intentionally avoids implying physical distinctions at CDC boundaries with thresholds to maintain methodological consistency and not imply more than we are able reliably report.

Furthermore, our study specifically targeted 'DNA-bound nucleosomes', tracking movements under 400nm across 20ms frames. A consequence of this approach is that typical nucleosome movement is confined within the range of one voxel, significantly limiting the detectability of transitions between CDCs in our experiments. For instance, considering a particle undergoing Brownian motion with a diffusion coefficient of $0.02 \mu\text{m}^2/\text{sec}$, over 99.9% of its displacements within a 20ms interval would fall below our lateral voxel size of 110nm. Regardless of these limitations, given the fractal nature of chromatin, if we were to be able to make such measurements, we would expect any changes in behavior as a molecule moves between different chromatin density regions to be gradual.

2. Were the fractal dimensions of each chromatin density class measured post-drug treatment? It would be nice to determine if the fractal dimensions estimated from cells treated with drugs can predict nucleosome diffusivity after drugs. If not, measuring fractal dimensions post-drug treatment would be beneficial for validating the model's accuracy and understanding the drug's impact on chromatin structure.

We appreciate the reviewer's suggestion and apologize if this wasn't clearly presented in the original submission. We have indeed measured the fractal dimensions of each chromatin density class post-drug treatment, as presented in Figures 4H and S7H of the original manuscript. These measurements from drug-treated cells were subsequently incorporated as parameters in our biophysical modeling for each condition. The accuracy and predictive power of the model, particularly in relation to nucleosome diffusivity post-drug treatment, were analyzed and discussed in Figures 5 and S10 of the original manuscript. These results, in addition to new results comparing the effect of drug treatment on the fractal dimension at the nuclear interior and periphery are also included in updated figures within the revised manuscript.

3. The authors should provide further details/explanation as to why different microscopes were used for imaging in live versus fixed cells. This will help others with future methodological and/or experimental design.

In choosing between HiST and LLSM microscopes for our imaging, we considered several factors relevant to our experimental design:

1. Photo-toxicity: Minimizing photo-toxicity is a critical consideration in our single molecule live cell imaging, such as the SPT experiments. This consideration is crucial not only for the health of the live sample but also reduces bleaching of the fluorescent probes. LLSM offers a high fluorescent signal while maintaining lower light dosage than HiST. This reduces phototoxicity and allows us to construct extended single nucleosome trajectories, enhancing the reliability of our downstream analysis.
2. Field of View: As the camera is orientated parallel to the coverslip, the HiST microscope offers a significantly larger field of view over a 2D plane compared to the LLSM with its oblique objective geometry. This feature is particularly beneficial for experiments such as our fixed-cell SMLM imaging to quantify the fractal dimension. In this approach, we acquire 100,000 frames. HiST imaging enabled us to capture multiple nuclei in a single imaging session, greatly enhancing the efficiency and scope of our data collection. Since the cells are fixed, increased phototoxicity is no longer a concern.

Reviewer #3 (Remarks to the Author):

The manuscript entitled “Correlative single molecule lattice light sheet imaging reveals the dynamic relationship between nucleosomes and the local chromatin environment” is devoted to developing an imaging platform to simultaneously visualize single protein dynamics together with the local chromatin environment in live cells; and to better determine the relationship between chromatin density, nucleosome motion, the physical properties of the interchromatin space, and specific nuclear functions.

The authors claim in the conclusion (rows 490-497), that this proposed imaging method serves as a foundation “in future studies to explore the relationship between chromatin density, nuclear function, and the diffusion of a diverse spectrum of nuclear proteins”.

The paper is very well written and structured in all sections. The topic would be a novel subject, to find NEWER biophysical properties and biological processes responsible for the observed differences in nucleosome motion, besides what is already known in this field. Unfortunately, the authors have not stated in detail how their work adds to the work mentioned below, with arguments, as to why their findings are significant. The imaging approach using the lattice light-sheet system is new, but the authors goal to create a manuscript with newer findings has not been fulfilled. Therefore, the results don't represent a sufficiently striking advance (noteworthy results) for publication in Nature Communications. So, I would not recommend the manuscript for this journal. It could be a very good paper for cellular biochemistry or other cellular and molecular journals.

We thank the reviewer for their careful reading and critical analysis of our work. We also appreciate that the reviewer did not have any technical or logical concerns about the results or conclusions in the manuscript. Although we disagree with the reviewer's opinion that we have failed to provide “new findings”, we do appreciate their perspective. Their critique has motivated us to add additional analysis to this revised manuscript delineating the relationship between nucleosome motion, chromatin density, and nuclear location and to state in more detail why our findings are either new or help to resolve conflicting results within the field.

I will explain to the authors the major weaknesses of their manuscript, so that the authors can understand the basis for my decision.

1. I don't have comments on the originality, data analysis, interpretation, enough detail is provided in the methods for the work to be reproduced, as all these parts are clear.

Thank you for this comment.

2. The paragraph (rows 60-64) starts to build the storyline of this paper, using two manuscripts (defined by the authors as ‘recent work’), one from 2022 (ref # 19) and the other one from 2009 (ref. #20), ignoring the fact that there are many other findings in these topics between the two references. Because the results/findings of the authors replicate already published existing results, the results of this manuscript are very observational for a Nature journal.

We appreciate the reviewer's feedback. We have revised the section in question to reflect the timeline of the referenced studies more accurately, ensuring that reference #20 is not characterized as recent.

While we made a concerted effort to acknowledge and cite the key research that has informed our work, we thank the reviewer for indicating additional studies and perspectives, many of which are now included as references in our manuscript.

We are unclear what the reviewer considers to be “very observational” results. While we discuss general principles of chromatin organization, we believe our study goes beyond mere observation. It offers a highly quantitative and statistically validated analysis, that in our view, contributes substantial and precise data to the field, which we consider a significant advancement for a publication in a journal of Nature Communication’s caliber. We comment below on why we believe our findings are novel and how they build upon prior work in the field.

3. Based on the statement below, the authors claim that” We hypothesized that the observed differences in nucleosome motion across CDCs may arise from various factors, including variations in nucleosome spatial packing, differences in the local viscoelastic properties of the material surrounding chromatin, and/or differences in active processes that may be localized to a given CDC. These factors, either individually or in combination, could contribute to the distinct patterns of nucleosome dynamics observed in different CDCs. Subsequently, our investigation aimed to identify and characterize the biophysical properties and biological processes responsible for these observed differences in nucleosome motion.”

Below we describe our rationale. In general, we feel that it is important to be precise about what prior manuscripts have shown via direct experimental measurements vs. indirect speculation and also to highlight when other manuscripts or data provide conflicting results to what the reviewer has indicated are known facts in the field. Overall, we thank the reviewer for pushing us more thoroughly highlight the novelty of our work relative to the field. This has led us in the revised manuscript to more thoroughly disambiguate the impact of nuclear location (e.g. interior vs periphery) vs. chromatin density. As a result of this, we have added the following additional findings those of the prior submission:

- 1) Surprisingly, we find that nucleosome motion at the periphery is independent of chromatin density whereas nucleosome motion at the interior is anticorrelated (Figure 3F in the revised manuscript). This was unexpected given that, as the reviewer notes below, the previous assumption was that “the decreased nucleosome motion is due to increased heterochromatin at the nuclear periphery forming Lamina-associated domains”. In the revised manuscript, we now show that this is not the case. Nucleosome motion at the periphery is equally constrained in both sparse euchromatic regions and dense heterochromatin environments suggesting that other factors beyond chromatin density (such as the previously mentioned LADs) govern nucleosome motion independently of heterochromatin vs. euchromatin.
- 2) We also find a distinct difference in the diffusive properties of Halotag at the nuclear periphery vs. at the interior. Surprisingly, despite there being increased heterochromatin at the periphery on average, halotag molecules appear to move faster in these regions than in the interior (Figure 3H in the revised manuscript). We discuss the implications for this (e.g. more efficient transport to/from the nuclear interior) in the revised discussion.
- 3) We build upon the biophysical model presented in Shinkai et al., by fully parameterizing it with empirical data and combining it with perturbation studies, none of which had been done previously. Together, we believe that this enhances the current understanding of the biophysical properties underpinning chromatin organization in cells

Below, we first address the reviewer's conclusions and then provide a point-by-point discussion of how our interpretation of the prior manuscripts differ from those of the reviewer.

Conclusion:

Known facts:

- the nucleosome motion is negatively correlated with local chromatin density
- the decreased nucleosome motion is due to increased heterochromatin at the nuclear periphery, forming Lamina-associated domains

We acknowledge that the negative correlation between nucleosome motion and local chromatin density is a well-supported conclusion, as also reflected in our measurements and discussed in our original manuscript. However, in contrast to this being a "known fact" we believe the finding to be more nuanced, first due to the existence of contradictory findings which we highlighted in the original submission:

"However, recent reports have also reported that nucleosome motion is independent of chromatin density, on average, when investigating replication inhibited cells with the same chromatin content, but double the nuclear volume compared to control cells (Iida S. et al. Science Advances 2023)"

And second, because many of the prior studies did not directly measure nucleosome motion relative to chromatin density, but rather compared nucleosome motion at the interior vs. at the periphery (Nozaki et al. Molecular Cell 2017 and Nagashima et al. JCB 2019) or selectively compared nucleosome motion in a binary fashion between low density regions and very high density chromocenters (Nozaki et al. Molecular Cell 2017 and Gomez-Garcia et al. Cell Reports 2021). Our study advances these findings by directly correlating nucleosome dynamics with chromatin density in real time. Furthermore, our approach goes beyond the traditional overgeneralizations of nuclear periphery versus interior or sparse chromatin vs. dense chromocenters, thus offering a more detailed analysis of nucleosome dynamics and organization within specific chromatin environments.

In our original submission, we observed that nucleosomes at the nuclear periphery exhibit not just slower, but also more anomalous diffusion compared to those in the nuclear interior. This anomalous diffusion was uncorrelated with chromatin density and suggested that properties beyond chromatin density alone influence nucleosome dynamics at the nuclear periphery. As stated above, our revised submission further emphasizes the distinct properties of the nuclear periphery in contrast to the nuclear interior, and includes an extended analysis comparing regions with similar chromatin densities in both areas. This comparative analysis reinforces our initial findings: the correlations between nucleosome dynamics/organization and local chromatin environment distinctly differ at the nuclear periphery compared to the nuclear interior. For example, our results reveal that nucleosome dynamics at the nuclear periphery are uniformly slower and do not correlate with local chromatin density. These observations challenge the assumption that slower nucleosome dynamics at the nuclear periphery are due to increased chromatin density and highlight the utility of disambiguating between nuclear location and chromatin density which is enabled by our imaging pipeline.

- the fractal dimension of nucleosome packing was also positively correlated with chromatin density
- the nucleosome packing is more structured and clustered in sparse chromatin regions and becomes more optimally space filling and randomly organized in dense chromatin environments

We recognize that a positive correlation between the fractal dimension of nucleosome packing and chromatin density has been proposed in prior research papers. However, we contend that this is commonly indirectly inferred from correlations rather than direct measurements and, as above, we contend that this is more nuanced than to be considered a “known fact”. For example, in Li, Y., Agrawal, V., Virk, R.K.A. et al. Analysis of three-dimensional chromatin packing domains by chromatin scanning transmission electron microscopy (ChromSTEM). *Sci Rep* 12, 12198 (2022), which the reviewer list below, we highlight that there are inconsistencies to the reviewer’s statement above. In Figure 5 (reproduced below), Li et al. present data showing that, on average, both radial chromatin mass density and fractal dimension (Dlog) are higher at the periphery than at the interior.

Figure 5

Comparing chromatin organization at nuclear periphery and interior. **(A)** Radial chromatin density was determined using ChromTEM images as a function of distance from the nuclear periphery. **(B)** Illustration showing how the mean radial chromatin mass density in **(A)** was estimated within 25 nm bands from the nuclear periphery to the center. Scale bar, 2 μm **(C)** ChromTEM image of the whole nucleus was utilized to segment chromatin at the nuclear periphery (red), and the interior (yellow). **(D)** The average ACF of chromatin mass density for the whole nucleus (blue), periphery (red), and interior (yellow) in the log-log scale. D was measured inside the domain regime (25 to 60 nm) by a linear regression fit of the ACF on a log-log scale. **(E)** A significantly higher D in the periphery compared to interior regions and the whole nucleus was observed. Each color represents a BJ cell nucleus (N = 29 cells, p value < 0.0001). The median D_{\log} (IQR) for the nuclear interior is 2.30 (2.11–2.53), the whole nucleus is 2.46 (2.35–2.67), and that for the periphery is 2.64 (2.53–2.76).

We agree that this supports the reviewers comment, but note that Li et al. do not directly measure the relationship between mass density and fractal dimension, but infer causation because both trends are correlated relative to the nuclear periphery. However, in the same manuscript, Li et al. reach the

opposite conclusion (Figure 3, E and F) when examining the relationship between fractal dimension and chromatin density relative to chromatin domains within the interior.

Here, they show that as radial chromatin density decreases away from the center of a domain, “the smaller length scales have a packing scaling $D < 3$ and that, as r increases up to around 100 nm, there is a sharp transition to the supra-domain regime with $= 3$.” Thus in Figure 3 and associated text, Li et al show that the fractal dimension of nucleosome packing is *negatively* correlated with chromatin density and that nucleosome packing is *less* structured and clustered ($D=3$) at the sparse chromatin region and becomes *less* optimally space filling ($D<3$) at the dense center of the domain.

Additionally, in prior work, Bancaud et al. 2009 inferred fractal dimension based on the diffusion of inert probes and concluded that euchromatin had a higher fractal dimension than heterochromatin (table 1 from this paper is reproduced below):

Table I Chromatin structural parameters are measured by ^aSPT, ^bFCS, ^cfractal kinetics in PA experiments, and ^{d,e,f}polymer physics considerations, fractal structural properties and rheological considerations, respectively (data)

	Euchromatin Heterochromatin	
Anomalous diffusion coefficient (γ)	0.73 ^a /0.79 ^b	0.77 ± 0.03 ^b
Fractal dimension (f)	2.6 ± 0.2 ^{a,c}	2.2 ± 0.2 ^c
Upper limit of self similarity (H)	100 nm	60 nm ^d
Lower limit of self similarity (h)	3 nm	3 nm
Fractal exponent (ε)	<0.01 ^c	0.18 ± 0.05 ^c
Accessible to total volume ratio (Φ)	0.39 ± 0.05 ^f	0.25 ± 0.04 ^e
	0.24 ± 0.04 ^f	0.11 ± 0.06 ^e
Diffusive hindrance (D/D_0)	0.3 ± 0.03 ^b	0.17 ± 0.03 ^b

In summary, based on our interpretation of the literature, the reviewer's statement has not yet been established as a "known fact" that precludes further study. Rather, we believe that better understanding the nuances of how chromatin density and nucleosome packing are related represents an important challenge for the field that highlights the value of parallel efforts and alternative approaches.

- the diffusion of free molecules increases in chromatin less dense regions

In our original submission, we presented data that actually contradicts this statement. Specifically, our findings indicate that a Halo Tag fused with a nuclear localization signal exhibits similar diffusion properties in both sparse and dense chromatin areas. Our data agree with prior studies (Bancaud et al. EMBO J 2009) that showed that, after correcting for volume exclusion effects, molecules display nearly identical recovery traces in both heterochromatin and euchromatin (c.f. reproduced Figure 2 from Bancaud below). Which the authors of this study interpreted as, "after steady-state normalization, all uptake kinetics seemed very similar at this temporal resolution (Figure 2C, right graphs)".

Figure 2 The nuclear rheology is heterogeneous. (A) NRK cell transiently expressing H2b-mRFP and mEGFP-5 were subjected to FCS measurements. Crosses on the H2b image indicate positions at which measurements were performed. The graph shows normalized auto correlation functions (ACF) obtained in the nucleoplasm (red) and the nucleolus (orange). Fits were performed with an anomalous diffusion model (solid curves), and we deduced residence times of 1050 and 3650 μs , and anomalous coefficients of 0.78 and 0.65 in nucleoplasm and nucleolus, respectively. The inset shows count rates, that is, intensities measured by FCS in the nucleoplasm (red) and nucleolus (orange). (B) Similar experiments performed with NIH3T3 cells. The green cross indicates the position of heterochromatin measurements, which was always quality controlled taking advantage of H2B-mRFP bleaching during FCS (Supplementary Figure S2). Graphs show normalized ACFs obtained in euchromatin (red), heterochromatin (green) and nucleoli (orange). Fits (solid curves) show more pronounced diffusion slow down in nucleoli than in heterochromatin, as inferred from the mEGFP-5 residence times of 3570 μs ($\alpha = 0.70$) and 1410 μs ($\alpha = 0.80$) in nucleoli and heterochromatin, respectively, in comparison to 790 μs ($\alpha = 0.77$) in euchromatin (bottom). Inset shows count rates measured in euchromatin (red), heterochromatin (green) and nucleoli (orange). (C) Selected frames of mPAGFP-2 half-nucleus PA time lapse imaging with the photoactivated region represented by the polygon on the pre-activation image. To visualize entry kinetics within nuclear compartments, 1.2- μm confocal slices were grabbed. High-quality images of mPAGFP-2 steady state and Hoechst distribution were acquired 60 s after PA (lower panel). Rings on the steady-state image correspond to regions in which the intensity redistribution was measured over time. Graphs at the left compare nucleolar (orange) and heterochromatin (green) fluorescence intensity measured over time in the regions highlighted with the corresponding coloured circles in the steady-state image to the intensity in a neighbouring nucleoplasmic area (red and purple regions). Graphs at the right display the same curves after steady-state renormalization. Scale bars 10 μm .

Both our data and Bancaud's data are also in agreement with Imai et al. MBOC 2017 that used orientation-independent-DIC microscopy to show that the material densities of euchromatic and heterochromatic regions are very similar. Finally, as we describe in more detail below, we do not believe that the supporting reference provided by the reviewer (Mazzocca, M et al. Nat Comms 2023) presents data that conclusively shows that free diffusing halotag moves slower in dense chromatin regions.

These hypotheses and findings were already covered in various published work, as follows:

- A) chromatin exhibits a polymeric fractal-like behavior and a radially decreasing mass-density from the center to the periphery.
- B) for the majority of the cells the Dlog at the periphery was again higher than Dlog at the non-periphery

Li, Y., Agrawal, V., Virk, R.K.A. et al. Analysis of three-dimensional chromatin packing domains by chromatin scanning transmission electron microscopy (ChromSTEM). Sci Rep 12, 12198 (2022). <https://doi.org/10.1038/s41598-022-16028-2>

We discussed the findings from this paper in more depth above. From our data, we find that on-average, the Dlog at the periphery was higher than Dlog at the non-periphery. We agree with the reviewer that this is likely because chromatin at the periphery has a higher relative percentage of high-density chromatin compared to the interior (c.f. Figure S6G of the revised submission). Surprisingly though, if we separately investigate the fractal dimension between comparable chromatin density regions at the interior vs. at the periphery, we find that, excepting class 1, there is actually a subtle trend of lower fractal dimension for similar density chromatin domains at the periphery than at the interior (c.f. Figure 3J of the revised submission).

C) “more compact chromatin located remote from IC channels can be attributed to the INC. It is enriched in epigenetic marks for low or silent transcriptional activity”

Popken J, Brero A, Koehler D, Schmid VJ, Strauss A, Wuensch A, Guengoer T, Graf A, Krebs S, Blum H, Zakhartchenko V, Wolf E, Cremer T. Reprogramming of fibroblast nuclei in cloned bovine embryos involves major structural remodeling with both striking similarities and differences to nuclear phenotypes of in vitro fertilized embryos. *Nucleus*. 2014;5(6):555-89. doi: 10.4161/19491034.2014.979712. PMID: 25482066; PMCID: PMC4615760.

We thank the reviewer for drawing attention to the seminal research on chromatin organization during the reprogramming of fibroblast nuclei. This study, along with others that perform correlative imaging of nuclear regulatory machinery and DNA in fixed cells, indeed serves as a cornerstone for our manuscript. Our analysis of the increased nucleosome motion in sparser chromatin regions, especially following transcription inhibition, is informed by these studies in fixed cells as well as pioneering work in single nucleosome imaging in live cells. However, our research takes a step further by moving beyond the correlative imaging of transcriptional regulators in fixed cells. We demonstrate the impact of these active biological processes on shaping the physical landscape of the perichromatin space, providing new insights into the dynamic interplay between nuclear organization and function.

Cremer T, Cremer M, Hübner B, Silahatoglu A, Hendzel M, Lanctôt C, Strickfaden H, Cremer C. The Interchromatin Compartment Participates in the Structural and Functional Organization of the Cell Nucleus. *Bioessays*. 2020 Feb;42(2):e1900132. doi: 10.1002/bies.201900132. PMID: 31994771.

We thank the reviewer for bringing our attention to this insightful perspective. Again, we note that our interpretations were informed by previous research suggesting a functional role for nuclear organization in shaping physiological processes.

D) Figure 6, the same conclusion as:

We thank the reviewer for highlighting these manuscripts. We did not mean to suggest that Figure 6 was a completely novel summary of nuclear structure. Rather, our goal with Figure 6 was to summarize an emerging consensus from many works in the field and to highlight how our new findings add to this picture. In the revised manuscript, we highlight more clearly how our work adds to this concept by: 1) disambiguating the roles of nuclear location vs. chromatin density on nucleosome motion, 2) providing direct measurements that inhibiting transcription preferentially destabilizes nucleosome motion in sparse chromatin regions (where transcriptional bursting also occurs), and 3) fully parameterizing the polymer model proposed in Shinkai et al with experimental data and describing where it does vs. doesn't explain the observed nucleosome motion.

Márton Gelléri, Shih-Ya Chen, Barbara Hübner, Jan Neumann, Ole Kröger, Filip Sadlo, Jorg Imhoff, Michael J. Hendzel, Marion Cremer, Thomas Cremer, Hilmar Strickfaden, Christoph Cremer, "True-to-scale DNA-density maps correlate with major accessibility differences between active and inactive chromatin", *Cell Reports*, Volume 42, Issue 6, 2023, 112567, ISSN 2211-1247, <https://doi.org/10.1016/j.celrep.2023.112567>.

This paper reveals that there is a > 60 fold density difference in DNA density between sparse/dense region of nucleus, which the authors propose will result in the limited accessibility of nanobeads with the size of 20-40 nm into dense DNA region. The final figure reproduces a portion of the summary figure from Miron et al. highlighting that dense repressed cores of chromatin domains are depleted for "active" transcription-associated proteins. The authors propose that

"DNA targets located in the interior of such ensembles may still be accessible for small macromolecules, such as small gene-specific transcription factors, but inaccessible for larger protein complexes (examples of estimated sizes are provided in Maeshima et al.¹⁹)."

Our findings with free-diffusing Halotag, support the contention that there is no diffusive barrier for small molecules into dense chromatin domains (also supported by the discussion of Bancaud et al. above). However, we believe that it is less clear whether larger protein complexes, which opposed to nanobeads, can dynamically assemble and disassemble would be excluded from dense domains. We note that Mazzocca et al cited below did not find a relationship between protein size and enrichment in any given density class, stating:

"We next checked if these differences in inaccessibility to DNA-dense nuclear regions could be explained by simple size-exclusion effects that would predict larger factors to be more excluded from CD regions. However, we observed no correlation between protein molecular weight and enrichment in higher DNA classes (Fig. 1e)—even when factors oligomerization was taken into account (Supplementary Fig. 2b)—suggesting that additional mechanisms might be at play."

Shinkai S, Nozaki T, Maeshima K, Togashi Y (2016) Dynamic Nucleosome Movement Provides Structural Information of Topological Chromatin Domains in Living Human Cells. *PLOS Computational Biology* 12(10): e1005136. <https://doi.org/10.1371/journal.pcbi.1005136>

As stated in our manuscript, we applied the model proposed in this paper to explain our experimental data. The model assumes chromatin follows a fractally organized Rouse polymer diffusing in viscoelastic medium. However, the authors of this original paper provided limited experimental validation of this model, concluding that slower than nucleosome motion at the nuclear periphery compared to the nuclear interior suggests that the chromatin domains are smaller at the nuclear periphery. This validation was performed by constraining the nucleosome diffusion coefficient and anomalous alpha exponent at the interior and periphery based on experimental measurements and then then varying the other model parameters to examine the how the model scales when tuning these unknown variables. We found that when the proposed model is fully parameterized with empirical measurements, it can explain the connection between nucleosome motion, fractal dimension, and the viscoelastic properties in high chromatin density regions but not in low chromatin density regions. We as well parameterized this model with empirical measurements from cells in which transcription has been pharmacologically inhibited to reveal that transcription constrains nucleosome motion in sparse chromatin regions. All of these are novel additions compared to the original manuscript referenced above.

Bizhanova A, Kaufman PD. Close to the edge: Heterochromatin at the nucleolar and nuclear peripheries. *Biochim Biophys Acta Gene Regul Mech.* 2021 Jan;1864(1):194666. doi: 10.1016/j.bbagr.2020.194666. Epub 2020 Dec 8. PMID: 33307247; PMCID: PMC7855492.

This paper reviewed the spatial organization of heterochromatin in mammalian cells, especially the ones that localize to the nuclear and nucleolar peripheries. It discussed its role in regulating gene transcription. We want to emphasize that we do not claim the novelty of heterochromatin's role in silencing genes. In this manuscript, we now confirm that, on-average, dense chromatin regions are enriched at the nuclear periphery. However, we also extend these findings to show that the nuclear periphery still contains very heterogeneous chromatin densities and that nucleosome motion in this region is in-fact independent of chromatin density.

Bajpai G, Amiad Pavlov D, Lorber D, Volk T, Safran S. Mesoscale phase separation of chromatin in the nucleus. *Elife*. 2021 May 4;10:e63976. doi: 10.7554/eLife.63976. PMID: 33942717; PMCID: PMC8139833.

In this paper, the authors simulated a polymer chain and studied the role of chromatin self-interaction and chromatin-lamina binding in shaping the overall distribution of chromatin. The paper reproduced separated heterochromatin and euchromatin domains and showed fractal organization in those chromatin domains. They revealed that upon certain tunable parameters, model can produce qualitatively similar distributions of chromatin compared to fixed and stained images. However, this paper did not investigate the implications for nucleosome motion, transcription regulation, or the viscoelastic properties of the interchromatin space as we do here.

E) "fractal dimension depends on the local chromatin density"

Almassalha, L., Tiwari, A., Ruhoff, P. et al. The Global Relationship between Chromatin Physical Topology, Fractal Structure, and Gene Expression. *Sci Rep* 7, 41061 (2017). <https://doi.org/10.1038/srep41061>

This paper models the relationship between the fractal dimension and the variance of chromatin density *fluctuations* (Fig 1G and Supplementary Equations 1). We note that this is fundamentally different than what the reviewer describes above as "local chromatin density". Further, this paper does not provide experimental measurements, but rather a plot of the model prediction. Finally, we comment that the findings from the model in this paper appear contradictory to what we experimentally observed here as higher fractal dimension leads to more optimal packing and fewer density fluctuations (e.g. when fractal dimension = 3, the distribution is uniform and therefore no density fluctuation will be observed).

We actually feel that Li Y. et al. 2022 (also from the Backman group) is a better reference for this statement. We discussed above how this paper found conflicting results regarding fractal dimension and chromatin density in different portions of the paper leading to our view that this relationship is less straightforward than the reviewer suggests.

Récamier V, Izeddin I, Bosanac L, Dahan M, Proux F, Darzacq X. Single cell correlation fractal dimension of chromatin: a framework to interpret 3D single molecule super-resolution. *Nucleus*. 2014 Jan-Feb;5(1):75-84. doi: 10.4161/nucl.28227. Epub 2014 Feb 19. PMID: 24637833; PMCID: PMC4028358.

This paper reports on the fractal organization of chromatin using single-molecule localization microscopy (similar to what we do in this manuscript). However, contrary to the reviewer's claim, this paper only provides a single measure for the whole cell. It does not investigate any relationship between fractal dimension and chromatin density. Our work builds upon these findings by differentiating the fractal dimension associated with different chromatin densities and at different locations within the nucleus.

F) “there are a variety of techniques to measure chromatin accessibility and related quantities using chemical probes”

Mansidor AR, Risca VI. Chromatin accessibility: methods, mechanisms, and biological insights. *Nucleus*. 2022 Dec;13(1):236-276. doi: 10.1080/19491034.2022.2143106. PMID: 36404679; PMCID: PMC9683059.

Tsompana, M., Buck, M.J. Chromatin accessibility: a window into the genome. *Epigenetics & Chromatin* 7, 33 (2014). <https://doi.org/10.1186/1756-8935-7-33>

We agree with the reviewer that there are a variety of techniques to measure chromatin accessibility. We also thank the reviewer for highlighting these review papers on biochemical and sequencing-based methods for examining chromatin accessibility. Indeed, we previously developed one of the methods described in Mansidor et al (ATAC-PALM) to directly visualize accessible chromatin regions in cells. However, we note that our current manuscript does not make any claim to have invented a novel method to measure chromatin accessibility. Rather, we use freely diffusing probes to quantitatively measure the viscoelastic properties of the interchromatin space. We are thus unclear how the existence of these methods might somehow undermine the novelty of this new work.

G) “biological processes such as gene transcription play a larger role in regulating nucleosome motion in chromatin sparse regions than chromatin dense regions”

Michael Babokhov, Kayo Hibino, Yuji Itoh, Kazuhiro Maeshima, “Local Chromatin Motion and Transcription”, *Journal of Molecular Biology*, Volume 432, Issue 3, 2020, Pages 694-700, ISSN 0022-2836, <https://doi.org/10.1016/j.jmb.2019.10.018>.

We are grateful to the reviewer for pointing out this influential review. In fact, we believe this review effectively sets the stage for our manuscript. The authors of this perspective highlight the emerging model in which transcription locally constrains chromatin motion, *potentially* via hubs formed in in sparse and intermediate chromatin regions based on prior live-cell nucleosome tracking papers comparing the nuclear interior vs. the nuclear periphery. But they also acknowledge that this concept has not been directly tested, specifically noting towards the end of this perspective: “It would be intriguing to see whether chromatin around the transcription hub is constrained.” Here, we provide experimental evidence for this by localizing the effects of transcription perturbation on nucleosome dynamics to sparse and intermediate chromatin classes, which had not been done before, and showing that this is also where transcriptional bursting events occur. This paves the way for future studies to simultaneously investigate hub-formation, transcriptional bursting, and nucleosome motion which will be an interesting area of investigation.

H) “transcription stabilizes nucleosome motion, but in some cases could destabilize the motion, it depends on the specific position and orientation of the binding motifs relative to the nucleosome”

Ekaterina Morgunova, Jussi Taipale, “Structural insights into the interaction between transcription factors and the nucleosome”, *Current Opinion in Structural Biology*, Volume 71, 2021, Pages 171-179, ISSN 0959-440X, <https://doi.org/10.1016/j.sbi.2021.06.016>.

We are grateful to the reviewer for directing us to this perspective on the structural interactions between chromatin and trans-regulatory machinery. While the review neither directly supports nor contradicts our findings, it does raise intriguing questions about how these interactions might stabilize or destabilize nucleosome motion. In our study, we observed that the degradation of the major subunit of RNA polymerase II, either through treatment with α -Amanitin or via conditional knockout, results in an overall increase in nucleosome dynamics. This observation aligns with previous studies indicating that the knockdown of structural proteins like CTCF and Rad21 also leads to similar global effects. We believe that an interesting follow up study to the one currently under review could include more comprehensive investigation into how the removal of specific trans-regulatory factors influences nucleosome dynamics.

Lai WKM, Pugh BF. Understanding nucleosome dynamics and their links to gene expression and DNA replication. *Nat Rev Mol Cell Biol.* 2017 Sep;18(9):548-562. doi: 10.1038/nrm.2017.47. Epub 2017 May 24. PMID: 28537572; PMCID: PMC5831138.

We thank the reviewer for referring us to this review. We are unclear as to how this review supports or diminishes the findings in our study, though we speculate that nucleosome repositioning during active transcription should necessarily lead to an apparent destabilization of the motion. This in part is why our data, as well as the observations of multiple other groups highlighted in the Babokhov et al. perspective referenced in section G of the review, is surprising in that it indicates transcription, or possibly the aggregation of the transcriptional machinery, tends to constrain nucleosome motion.

l) “the diffusion of free molecules increases in chromatin less dense regions”

Mazzocca, M., Loffreda, A., Colombo, E. et al. Chromatin organization drives the search mechanism of nuclear factors. *Nat Commun* 14, 6433 (2023). <https://doi.org/10.1038/s41467-023-42133-5>

Upon a thorough examination of the cited paper by Mazzocca et al., our reading is that the authors do not explicitly state or provide data that directly supports the claim that “the diffusion of free molecules increases in less dense chromatin regions”. We’ve reproduced Figure 1g below which we believe most directly deals with this statement.

independent experiments, error bar: SD, statistical test: Pearson correlation). **g** Localization of NFs molecules in chromatin depending on their instantaneous diffusion coefficient ($n_{\text{cells}} = 31, 32, 29, 31, 32$, from two biologically independent experiments for HaloTag, p65, p53, CTCF, and Histone H2B, respectively). Source data are provided as a Source Data file.

In our interpretation of this figure, p53 (IR) and CTCF do show an enrichment of fast moving molecules at low density chromatin regions and slow moving molecules at high density chromatin (indicated by the

checkerboard like pattern in the heatmap), but these are not inert “free” molecules. In contrast p65 (TNF) shows an enrichment in low density chromatin regardless of diffusion speed. Our opinion is that HaloTag falls somewhere in between these extremes, but closer to the plot shown for p65. The slowest moving Halotag molecules (left most column) actually appear to be most enriched in the lowest density chromatin class (1) and second most enriched at the highest density class (4). Above an x-value of ~ 2 , it appears that all diffusional speeds are enriched in low density domains (similar to p65). We note that the presentation of data in this figure is complex and is challenging (at least to us) to interpret on first pass. Upon careful examination, we note that the data for these heatmaps are normalized “column-wise” such that the heatmap values for each column showing enrichment or depletion for a given diffusion speed sums to one. Because of this normalization, it is not possible to make comparisons within a given row of the heatmap. For example, it is not possible to look solely at the row for Halotag in chromatin density class 4 and conclude that there are a greater proportion of slow diffusing molecules than fast diffusing molecules in this class. This is because each column has been normalized independently and summing each row along the x-axis does not represent a proportion (as it does when summing along each column). In other words, the figure presents data showing that “given a specific diffusion coefficient, what is the relative enrichment in different chromatin density classes”. It does not allow one to conclude the inverse relationship of “given a specific chromatin density class, what is the distribution of diffusion coefficients”, which is what we show here in this submission.

We could not find in the manuscript where the authors state that that “diffusion of free molecules increases in chromatin less dense regions”. Rather our interpretation of this figure is consistent with what the authors do state in this manuscript which is that inert HaloTag tracers of all speeds tend to be 'excluded from CD regions and display isotropic diffusion'. This, combined with conflicting results from Bancaud et al. 2009 (described above), and our own data in this submission showing that the diffusion speed of free molecules is *independent* of chromatin density makes us hesitant to agree that the reviewers statement above is already established as a “known fact” in the field.

Overall, we value the reviewer’s expert insight and comments. We hope that the revised manuscript and this response more effectively present how we view the novelty of our contribution relative to the current state of the field.

Reviewers' Comments:

Reviewer #1:

Remarks to the Author:

The authors adequately addressed most of my comments and discussed their study's limitations. The revised manuscript has been much improved. I would support the publication of this work. Yet, I have a minor issue with the new Fig. 3E:

Fig. 3E shows the CDCs are segmented by the pixel level (~ 100 nm?) in the XY direction. I wonder whether the lateral resolution (837 nm) is not sufficient for the CDC segmentation. The authors may want to mention this limitation.

Reviewer #2:

Remarks to the Author:

My questions have been resolved. One small thing: I encourage the authors to update the methods section to include some or all of the discussion in their 'response to reviewers' that was related to choosing between HiST and LLSM microscopes for experiments. This will aid readers who wish to do similar experiments. This text may have already been included, in which case I apologize, but I wasn't able to find it when I did a quick scan.

Reviewer #3:

Remarks to the Author:

The authors addressed all questions and suggestions I've had. The paper is now recommended for the Nature Communication journal.

Reviewer #1 (Remarks to the Author):

The authors adequately addressed most of my comments and discussed their study's limitations. The revised manuscript has been much improved. I would support the publication of this work. Yet, I have a minor issue with the new Fig. 3E:

Fig. 3E shows the CDCs are segmented by the pixel level (~100 nm?) in the XY direction. I wonder whether the lateral resolution (837 nm) is not sufficient for the CDC segmentation. The authors may want to mention this limitation.

We appreciate the reviewer's comment. With regards to the concern about whether the resolution is "sufficient" for CDC segmentation, we wish to stress that we are not segmenting physical domains with a characteristic size. Rather, we have emphasized in lines 124-126 of the manuscript that the chromatin density classifications (CDCs) represent a statistical partitioning of a smooth intensity distribution, rather than describing physically distinct chromatin "domains". Therefore, the CDC segmentation is simply a measure of the underlying chromatin density at the resolution of the microscope which is reported as 334 x 837 nm resolution (full width half maximum) laterally and axially in lines 104 of the revised manuscript.

Reviewer #2 (Remarks to the Author):

My questions have been resolved. One small thing: I encourage the authors to update the methods section to include some or all of the discussion in their 'response to reviewers' that was related to choosing between HiST and LLSM microscopes for experiments. This will aid readers who wish to do similar experiments. This text may have already been included, in which case I apologize, but I wasn't able to find it when I did a quick scan.

We appreciate the reviewer's comment and have added further clarification to the manuscript on our choice of HiST microscopy for the fixed cell single molecule localization experiments. This clarification is in lines 198-200 of the updated manuscript:

"Because phototoxicity is not important for fixed-cell imaging, we chose HIST because of its larger field of view parallel to the coverslip compared to lattice light sheet microscopy."

Reviewer #3 (Remarks to the Author):

The authors addressed all questions and suggestions I've had. The paper is now recommended for the Nature Communication journal.

We thank the reviewer for their thorough review of our manuscript. We believe their suggestions have significantly improved the paper.